# Learning Manifold and Itô Dynamics with Branched ⌄ Neural Rough Differential Equations

Luke Thompson [*1]   Dai Shi [*2]   Lequan Lin [1]   Junbin Gao [1]   Andi Han [1]

## Abstract

Neural rough differential equations (NRDEs) stay accurate under irregular sampling while taking far fewer integration steps than standard neural differential equations, summarising a finely sampled driver by its log-signature and advancing the hidden state over coarse intervals using the log-ODE method. This efficiency rests on the shuffle algebra, the algebraic counterpart of Stratonovich calculus. This reliance means NRDEs cannot expose the quadratic-variation terms Itô dynamics require, nor the ordered covariant derivatives that govern Itô flows on connection-equipped manifolds. Ameliorating this, we introduce Branched Neural Rough Differential Equations (B-NRDEs), a Hopf-algebraic framework that recasts the NRDE log-ODE step as geometric numerical integration on the state-space manifold, matching the driving algebra to the governing calculus: Grossman–Larson rooted trees for Euclidean Itô dynamics, Munthe–Kaas–Wright planar rooted trees for ordered covariant derivatives on manifolds, and the shuffle algebra in the classical Stratonovich case. This yields intrinsic coarse-step dynamics that exactly preserve manifold constraints. Finally, we introduce a branched signature-kernel objective to enable Itô-consistent law matching by making quadratic variation terms visible during training. On rough Bergomi volatility, sim-to-real $SO(3)$ forecasting, and SPD covariance dynamics, B-NRDEs offer a unified, effective approach to stochastic and manifold-valued dynamics beyond the Euclidean-Stratonovich setting.

[1]University of Sydney, Australia [2]Cambridge University, United Kingdom. Correspondence to: Luke Thompson <luke-a-thompson@outlook.com>, Andi Han <andi.han@sydney.edu.au>.

*Proceedings of the 43rd International Conference on Machine Learning*, Seoul, South Korea. PMLR 306, 2026. Copyright 2026 by the author(s).

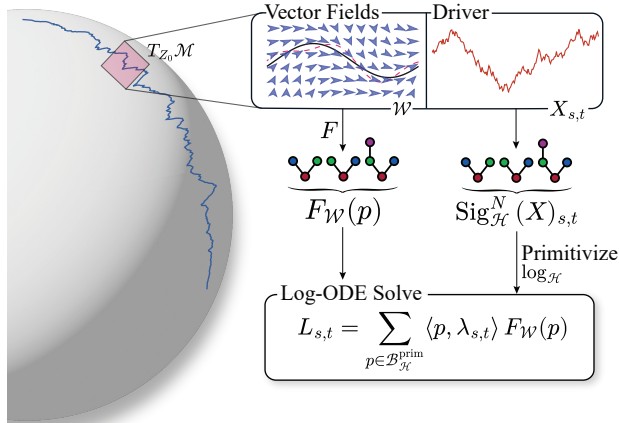

*Figure 1.* Illustration of the log-ODE method for branched neural rough differential equations (B-NRDEs). The control segment $X_{s,t}$ is lifted to $\mathbf{X}_{s,t}^{\mathcal{H}} \in \mathcal{H}$ for the chosen Hopf algebra $\mathcal{H} \in \{\mathcal{H}_{\sqcup\!\sqcup}, \mathcal{H}_{\mathrm{GL}}, \mathcal{H}_{\mathrm{MKW}}\}$, then transformed to a log-signature $\lambda_{s,t}$ in primitive coordinates $p \in \mathcal{B}_{\mathcal{H}}^{\mathrm{prim}}$. The tangent vector fields $\mathcal{W}$ are lifted through the pseudo bialgebra map $F_{\mathcal{W}}$ to matching primitive-indexed vector fields on the state manifold. These are combined into the log-ODE vector field $L_{s,t}$, whose ODE flow gives the local update from $s$ to $t$.

## 1. Introduction

Learning continuous-time dynamics from time series is a foundational problem in machine learning, with applications ranging from robotics (Duong & Atanasov, 2021; Chee et al., 2022) and molecular dynamics (Huang et al., 2023; Schreiner et al., 2023) to quantitative finance (Gierjatowicz et al., 2020; Issa et al., 2023). Neural controlled differential equations (NCDEs) (Kidger et al., 2020) approach this task by parameterising the vector field of a controlled differential equation with a neural network. To improve scalability and robustness to sampling rates, neural rough differential equations (NRDEs) (Morrill et al., 2021b) lift the control path to its log-signature, a series of iterated integrals, and solve the resulting system via the log-ODE method at coarser discretisations (Castell & Gaines, 1996).

However, geometric NRDEs are driven by geometric signatures whose coordinates satisfy the shuffle product identity. This is appropriate for Stratonovich integration, which preserves the usual chain rule, but not for Itô integration. Itô's

product rule generates quadratic-variation corrections, so products of Itô iterated integrals contain second-order terms that are not represented by the original tensor coordinates. This limits the use of geometric signatures in settings where adapted causal Itô modelling assumptions are required. On manifolds, the obstruction is sharper. Itô integration is defined relative to a connection, and its expansion involves higher covariant derivatives. Since these operators do not generally commute, the signature algebra must preserve their ordered, branched composition rather than identify them through shuffle relations.

To address these limitations, we introduce B-NRDEs, a unified framework for learning dynamics using the log-ODE method under Itô integration and over manifolds. By lifting the control path to a Hopf algebra of *rooted trees* — the Grossman–Larson algebra for Euclidean Itô calculus or the Munthe–Kaas–Wright algebra for manifolds, we derive a generalised NRDE that strictly respects the geometric and causal constraints of the domain. See Table 1 for comparisons of Hopf algebras and Figure 1 for an illustration of the proposed framework. Our **main contributions** are as follows:

1. We model path signatures in the Grossman–Larson Hopf algebra $\mathcal{H}_{\mathrm{GL}}$, whose rooted-tree basis represents Itô-type iterated integrals. We then introduce the first neural dynamics model utilising the Munthe–Kaas–Wright Hopf algebra $\mathcal{H}_{\mathrm{MKW}}$ of planar rooted trees to strictly enforce manifold constraints for Itô-type dynamical systems.

2. We unify the above approaches via *pseudo bialgebra maps* (Kern & Lyons, 2023), which convert Hopf algebraic structures to learned vector fields and differential operators on the target manifold. This yields a general log-ODE formulation for Stratonovich and Itô dynamics on connection-equipped manifolds. In this work, we instantiate the numerical solver on homogeneous spaces, with tangent vectors represented in frame or Lie-algebra coordinates and integrated through a group action.

3. We enable Itô-consistent neural dynamics training both in Euclidean space and on manifolds by developing a branched signature-kernel objective.

4. We release *Roughrax*, a JAX package for branched rough paths, and autodifferentiable numerical solution for rough differential equations (RDEs) over manifolds.

## 2. Preliminaries

In this section, we first review the standard formulation of Neural CDEs and RDEs to establish the Euclidean baseline (Section 2.1). We then discuss the machinery required to extend the Log-ODE framework to non-Euclidean and Itô

settings. We first introduce rough path Hopf algebras (Section 2.2), then present pseudo bialgebra maps that send algebra elements to differential operators on $\mathcal{M}$ (Section 2.3).

### 2.1. Neural Controlled and Rough Differential Equations

Let $\big((t_0, x_0), \ldots, (t_n, x_n)\big)$ be a potentially irregular time series with observations $x_i \in \mathbb{R}^d$. We construct a continuous interpolation $X : [t_0, t_n] \to \mathbb{R}^d$ such that $X_{t_i} = x_i$.

**Neural Controlled Differential Equations.** A NCDE (Kidger et al., 2020) evolves a hidden state $h_t \in \mathbb{R}^u$ driven by the path $X$. Let $\xi_\phi : \mathbb{R}^d \to \mathbb{R}^u$, $\ell_\psi : \mathbb{R}^u \to \mathbb{R}^w$ be an encoder and output network respectively, and $g_\theta : \mathbb{R}^u \to \mathbb{R}^{u \times d}$ be a vector field parametrisation. The forward pass is defined by the Riemann-Stieltjes integral:

$$h_t = h_{t_0} + \int_{t_0}^t g_\theta(h_s) \, \mathrm{d}X_s, \tag{1}$$

where initial hidden state $h_{t_0} = \xi_\phi(X_{t_0})$, and output $y_t = \ell_\psi(h_t)$, $g_\theta(h_s) \, \mathrm{d}X_s$ denotes matrix-vector multiplication. This formulation allows the latent state to react continuously to incoming data.

**Neural Rough Differential Equations.** While highly expressive, NCDEs are expensive for long sequences. NRDEs (Morrill et al., 2021b) reduce this cost by applying the log-ODE method: over each window $I_j = [t_j, t_{j+1}]$, the path $X$ is summarised by its log-signature $\lambda_j$. This log-signature determines an autonomous vector field, denoted $\tilde{g}_{\theta, \lambda_j}$, obtained by applying the log-ODE lift to the base neural vector field. The hidden state is then advanced by the ordinary differential equation (ODE)

$$h_t = h_{t_j} + \int_{t_j}^t \tilde{g}_{\theta, \lambda_j}(h_s) \, \mathrm{d}s, \qquad t \in I_j. \tag{2}$$

Here, $\lambda_j$ sets the coefficients of the ODE solved on $I_j$, rather than serving as a new path differential, yielding a higher-order approximation that permits larger step sizes.

### 2.2. Rough Path Hopf Algebras

Hopf algebras organise iterated-integral combinatorics and the product identities needed for computation. Table 1 summarises the regimes supported by each Hopf algebra.

**Definition 2.1** (Hopf algebra and primitives)**.** Let $\mathcal{H} = (H, \star, \eta, \Delta, \varepsilon, S)$ be a Hopf algebra on a vector space $H$ with product $\star : H \otimes H \to H$ and coproduct $\Delta : H \to H \otimes H$; we denote a basis of $H$ by $\mathcal{B}_\mathcal{H}$, and write $\mathrm{Prim}(\mathcal{H}) := \{p \in H : \Delta p = p \otimes 1 + 1 \otimes p\}$ with a chosen primitive basis $\mathcal{B}_\mathcal{H}^{\mathrm{prim}} \subset \mathrm{Prim}(\mathcal{H})$. We suppress $\eta$, $\varepsilon$ and $S$ below, since only $\star$ and $\Delta$ are immediately relevant to our constructions.

*Table 1.* Summary of the integration regimes encoded by each Hopf algebra and their first application to neural differential equations.

| Hopf algebra | Stratonovich | Euclidean Itô | Manifold Itô |
|---|---|---|---|
| $\mathcal{H}_{\sqcup\sqcup}$ (Morrill et al., 2021b) | ✓ | ✗ | ✗ |
| $\mathcal{H}_{\text{GL}}$ (this work) | ✗ | ✓ | ✗ |
| $\mathcal{H}_{\text{MKW}}$ (this work) | ✗ | ✓ | ✓ |

**Stratonovich (geometric) signatures and $\mathcal{H}_{\sqcup\sqcup}$.** For a $d$-dimensional control path $X = (X^{(1)}, \ldots, X^{(d)})$, the truncated geometric signature collects its iterated Stratonovich integrals in word coordinates indexed by $e_{i_1} \otimes \cdots \otimes e_{i_m}$. We use the standard tensor-algebra convention in which signatures are group-like, and log-signatures are computed using the tensor product; the shuffle product appears dually as the product identity satisfied by signature coordinate functions (Kidger et al., 2020; Morrill et al., 2021b). The defining feature of this setting is the Stratonovich product rule. At level two, we have

$$X^{(i)}_{s,t} X^{(j)}_{s,t} = \int_s^t X^{(i)}_{s,u} \circ \mathrm{d}X^{(j)}_u + \int_s^t X^{(j)}_{s,u} \circ \mathrm{d}X^{(i)}_u. \quad (3)$$

Algebraically, (3) is encoded by $e_i \sqcup\sqcup e_j = e_i \otimes e_j + e_j \otimes e_i$. Thus $\mathcal{H}_{\sqcup\sqcup}$ naturally represents geometric (Stratonovich) rough paths: iterated integrals are indexed by tensor coordinates and obey shuffle identities. Since Stratonovich integration is coordinate-free, the geometric signature remains appropriate for Stratonovich dynamics on manifolds.

**Itô (branched) signatures and $\mathcal{H}_{\text{GL}}$.** Under Itô integration, the product rule includes a quadratic-variation term:

$$X^{(i)}_{s,t} X^{(j)}_{s,t} = \int_s^t X^{(i)}_{s,u} \mathrm{d}X^{(j)}_u + \int_s^t X^{(j)}_{s,u} \mathrm{d}X^{(i)}_u + \langle X^{(i)}, X^{(j)} \rangle_{s,t}. \quad (4)$$

Over the original $d$-dimensional word coordinates, the correction term $\langle X^{(i)}, X^{(j)} \rangle$ is not exposed as an independent level-two driver coordinate. It can be represented only indirectly, for example, by augmenting the path through a lead–lag lift. Hence, the unaugmented shuffle representation is not the natural signature space for Itô modelling. We therefore work in the Hopf algebra of Grossman & Larson (1989), $\mathcal{H}_{\text{GL}}$, on rooted trees, which underpins branched rough paths (Gubinelli, 2010). Here, trees index the additional non-geometric iterated-integral coordinates, and in particular provide a (symmetric) second-order coordinate corresponding to the quadratic variation in (4).

**Itô (branched) signatures over manifolds and $\mathcal{H}_{\text{MKW}}$.** For dynamics on a manifold $\mathcal{M}$ with connection $\nabla$, the local flow involves *ordered* compositions of covariant derivatives: in general $\nabla_U \nabla_V \neq \nabla_V \nabla_U$, and the induced curvature terms depend on the order in which directions are applied.

This order-sensitivity is captured by the Munthe-Kaas & Wright (2006) Hopf algebra, $\mathcal{H}_{\text{MKW}}$, defined over *planar* rooted trees which fix a left-to-right child order at each node to index ordered iterated covariant derivatives. In contrast, non-planar rooted trees symmetrise child order and collapse the order-sensitive flow terms. See Appendix A for further algebraic details. We show a visual representation of the Stratonovich and Itô conventions in Figure 2.

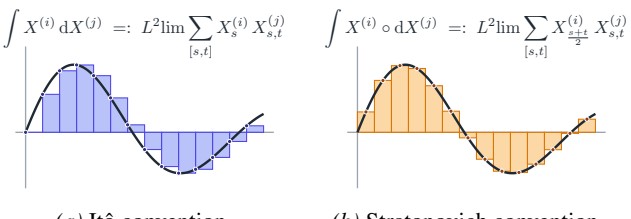

*(a)* Itô convention.      *(b)* Stratonovich convention.

*Figure 2.* Riemann–Stieltjes approximations under different evaluation conventions. Redrawn from Bellingeri et al. (2024).

### 2.3. Vector Fields and Pseudo Bialgebra Maps

Pseudo-bialgebra maps (Kern & Lyons, 2023) identify how signature coordinates act on functions through vector fields. Let $\mathcal{M}$ be a smooth manifold, let $\Gamma(T\mathcal{M})$ denote its smooth vector fields, and let $\text{Diff}(\mathcal{M})$ be the algebra of linear differential operators on $C^\infty(\mathcal{M})$. Given a connection $\nabla$ when covariant derivatives are required, a pseudo-bialgebra map sends each word or tree basis element of the chosen Hopf algebra to the corresponding differential operator on $\mathcal{M}$. Thus, the signature expansion and the induced flow expansion are evaluated in the same word- or tree-indexed coordinates.

**Definition 2.2** (pseudo bialgebra map (Kern & Lyons, 2023, Def. 4.1)). A linear map $F \colon \mathcal{H} \to \text{Diff}(\mathcal{M})$ is a pseudo bialgebra map if

$$F(\tau \star \sigma) = F(\tau) \circ F(\sigma)$$

and, for all $\tau \in \mathcal{H}$ and $\phi, \psi \in C^\infty(\mathcal{M})$, $m \circ (F \otimes F)(\Delta \tau)(\phi \otimes \psi) = F(\tau)(\phi\psi)$, where $m(a \otimes b) = ab$ denotes pointwise multiplication of functions.

Given a collection of smooth vector fields $\mathcal{W} = (W^{(1)}, \ldots, W^{(d)})$ on $\mathcal{M}$, the elementary differential associated with a rooted tree is defined recursively. For the single-node tree $F_{\mathcal{W}}(\bullet_i) = W^{(i)}$. For a rooted tree $\tau = [\tau_1 \ldots \tau_k]_i$, whose root has colour $i \in \{1, \ldots, d\}$ and $\tau_1, \ldots, \tau_k$ are its immediate child nodes, set

$$F_{\mathcal{W}}([\tau_1, \ldots, \tau_k]_i) = \nabla^k_{F_{\mathcal{W}}(\tau_1), \ldots, F_{\mathcal{W}}(\tau_k)} W^{(i)}. \quad (5)$$

Intuitively, we apply the $k$-th covariant derivative to the base vector field $W^{(i)}$, and ask how much it moves along the directions of vector fields specified by the subtrees $\tau_1, \ldots, \tau_k$. Crucially, if $\mathcal{H} = \mathcal{H}_{\text{MKW}}$, the order of arguments in the covariant derivative $\nabla^k$ corresponds to the planar order of subtrees, and the chain rule for covariant derivatives.

# 3. Methodology

We define B-NRDE by extending log-NCDE (Walker et al., 2024) to non-Euclidean geometries. The model learns an algebra-valued vector field and integrates the dynamics using the log-ODE method. Figure 1 and Algorithm 1 summarise the resulting piecewise log-ODE formulation.

---

**Algorithm 1** B-NRDE

---

**Input:** Hopf algebra $\mathcal{H} \in \{\mathcal{H}_{\sqcup\sqcup}, \mathcal{H}_{\mathrm{GL}}, \mathcal{H}_{\mathrm{MKW}}\}$, truncation depth $N$, partition $\{[t_k, t_{k+1}]\}_{k=0}^{M-1}$, path segments $X_{t_k, t_{k+1}}$, vector-field lift $F_W$, and initial condition $Y_0 \in \mathcal{M}$
**Precompute:** build the depth-$N$, $d$-decorated basis of $\mathcal{H}$, primitive basis $\mathcal{P} \leftarrow \mathcal{B}_{\mathcal{H}}^{\mathrm{prim}}$, signature routines, Hopf logarithm, and primitive-field evaluation schedules
**Cache offline:** for each segment $X_k$, compute

$$\lambda_k = \log_{\mathcal{H}}\big(\mathrm{Sig}_{\mathcal{H}}^N(X_k)\big) = \sum_{p \in \mathcal{P}} \lambda_k^p p.$$

**for** $k = 0$ **to** $M - 1$ **do**
   $L_k(Y) \leftarrow \sum_{p \in \mathcal{P}} \lambda_k^p F_W(p)(Y)$      ▷ lifted field
   $\dot{Z}_\tau = L_k(Z_\tau),\ Z_0 = Y_k;\ Y_{k+1} \leftarrow Z_1$  ▷ log-ODE step
**end for**
**Output:** trajectory samples $\{Y_k\}_{k=0}^M$

---

## 3.1. Signature Primitivisation

Given a path segment $X_{s,t}$, its $\mathcal{H}$-signature is the group-like element $\mathbb{X}_{s,t}^{\mathcal{H}} \in \mathcal{H}$ whose coordinates encode the corresponding iterated integrals, meaning that $\Delta \mathbb{X}_{s,t}^{\mathcal{H}} = \mathbb{X}_{s,t}^{\mathcal{H}} \otimes \mathbb{X}_{s,t}^{\mathcal{H}}$. The log-ODE method requires expressing the signature in primitive elements via the Hopf logarithm:

$$\log_{\mathcal{H}}(g) := \sum_{n \geq 1} \frac{(-1)^{n-1}}{n} (g - 1)^{\star n}, \qquad (6)$$

where $1$ is the unit and $\star$ is the product in $\mathcal{H}$. $\log_{\mathcal{H}}$ yields a primitive element, which we term the $\log_{\mathcal{H}}$-signature.

The product $\star$ of the working Hopf algebra fixes the meaning of (6) Throughout, $\mathcal{H}$ denotes the graded dual in which enhanced paths are group-like and logarithms primitive, with coordinate identities such as the shuffle product rule stated dually. For $\mathcal{H}_{\sqcup\sqcup}$, $\star$ is tensor concatenation, whose dual coordinate product is the shuffle; for $\mathcal{H}_{\mathrm{GL}}$ and $\mathcal{H}_{\mathrm{MKW}}$, $\star$ is the tree grafting product (Sections A.4.3 and A.4.5). A single Hopf logarithm thus yields the segment log-signatures $\{\lambda_j\}_{j=0}^{T-1}$ used by B-NRDE across all three regimes.

## 3.2. Neural Vector Fields

The learnable component of a B-NRDE is a collection of atomic driving vector fields, not the full primitive-indexed

log-ODE field. For $z \in \mathcal{M}$, we parameterise

$$W_\theta(z) = (W_\theta^{(1)}(z), \ldots, W_\theta^{(d)}(z)), \quad W_\theta^{(i)}(z) \in T_z\mathcal{M}.$$

Thus $\mathcal{W}_\theta = (W_\theta^{(1)}, \ldots, W_\theta^{(d)})$ contains one vector field per driver channel. B-NRDE learns only these atomic channel fields. The primitive-indexed fields used by the log-ODE are then generated deterministically from $W_\theta$ by the vector-field lifts described in the sequel.

The display above is *representation-free*: the method only requires that the network parameterisation determines an element of $T_z\mathcal{M}$ for each state $z$ and driver channel. In our experiments, we use a homogeneous space realisation, in which the network outputs frame coordinates and the numerical flow is applied through the corresponding group action. This enforces the manifold constraint at every solver substep, avoiding the ambient-space projection or retraction errors that arise when integrating extrinsically.

## 3.3. Vector Field Lifts

We implement the pseudo bialgebra map as a *vector field lift* $F_W : \mathcal{B}_{\mathcal{H}}^{\mathrm{prim}} \to \Gamma(TM)$ mapping primitive basis elements of the chosen Hopf algebra to vector fields on the state space. This follows from the coproduct Leibniz property of Definition 2.2, which ensures $F_{\mathcal{W}}(p) \in \Gamma(TM)$ for $p \in \mathcal{B}_{\mathcal{H}}^{\mathrm{prim}}$ (Kern & Lyons, 2023, Prop. 4.3); see Proposition B.5. In all cases, the implementation separates basis and combinatorial precomputation from auto-differentiation-based evaluation. The atomic fields are those defined in Section 3.2, and we write $W := \mathcal{W}_\theta$, $F_W := F_{\mathcal{W}_\theta}$ for notational simplicity.

**Primitive bases and combinatorics.** *Words* ($\mathcal{H}_{\sqcup\sqcup}$): We use the Lyndon basis of the free Lie algebra, indexed by Lyndon words $\ell = (i_1, \ldots, i_m)$ over $\{1, \ldots, d\}$ with $|\ell| \leq N$; see Definition A.2. We enumerate these words by degree using Duval's generator (Duval, 1988). For each non-letter Lyndon word, we store its standard factorisation

$$\ell = uv, \qquad v \text{ the longest Lyndon suffix}, \qquad (7)$$

as the pair of indices corresponding to $u$ and $v$.

*Trees* ($\mathcal{H}_{\mathrm{GL}}, \mathcal{H}_{\mathrm{MKW}}$): primitives are represented by decorated rooted trees, stored recursively as $(\tau_1, \ldots, \tau_k, c)$, where $c \in \{1, \ldots, d\}$ is the root colour and $\tau_1, \ldots, \tau_k$ are the root subtrees. We precompute the indices of these subtrees, giving the recursive evaluation schedule shared by both tree algebras. For $\mathcal{H}_{\mathrm{GL}}$, unordered rooted-tree shapes are enumerated in Beyer & Hedetniemi (1980) lexicographic order, with colourings identified under tree symmetries. For $\mathcal{H}_{\mathrm{MKW}}$, ordered rooted-tree shapes are generated recursively from ordered compositions of the remaining node count; the left-to-right order of children is part of the basis element, and all colourings are distinct.

**Shared differential-operator scheme.**

The lifted fields are defined intrinsically from vector fields on $\mathcal{M}$. For $\mathcal{H}_{\sqcup\sqcup}$, Lyndon primitives are evaluated using the ordinary Lie bracket of vector fields, while for $\mathcal{H}_{\mathrm{GL}}$ and $\mathcal{H}_{\mathrm{MKW}}$, tree primitives are evaluated using iterated covariant derivatives with respect to the chosen connection $\nabla$. In the numerical implementation, these intrinsic operations are evaluated by forward-mode automatic differentiation in the chosen coordinate or frame representation.

**Evaluation: shuffle primitives (bracket recursion).** Given atomic channel fields $W^{(1)}, \ldots, W^{(d)}$, we set $F_W((i)) = W^{(i)}$ for the one-letter word $(i)$.

$$F_W(w) = [F_W(u), F_W(v)].$$

**Evaluation: tree primitives (multi-ary node recursion).** Fix a coloured rooted tree $p = (\tau, c)$. For each node $v$ of $\tau$, let $C(v) = (u_1, \ldots, u_k)$ denote its ordered children (intrinsic for $\mathcal{H}_{\mathrm{MKW}}$; any fixed order for $\mathcal{H}_{\mathrm{GL}}$). We compute subtree fields $V_v$ in postorder. Leaves satisfy $V_v = W^{(c(v))}$. If $v$ has ordered children $C(v) = (u_1, \ldots, u_k)$, we set

$$V_v = \nabla^k_{V_{u_1}, \ldots, V_{u_k}} w_{c(v)}.$$

Here $\nabla^k$ is the iterated covariant derivative, evaluated recursively for arbitrary vector fields $U_1, \ldots, U_k, W \in \Gamma(T\mathcal{M})$:

$$\nabla^k_{U_1, \ldots, U_k} W = \nabla_{U_1} \left( \nabla^{k-1}_{U_2, \ldots, U_k} W \right)$$
$$- \sum_{j=2}^{k} \nabla^{k-1}_{U_2, \ldots, \nabla_{U_1} U_j, \ldots, U_k} W.$$

Finally, $F_W(p) = V_r$, for the root node $r$. This is the computational form of (5): each internal node is evaluated by JVP-based corrected covariant derivatives, and planarity in $\mathcal{H}_{\mathrm{MKW}}$ enters through the child order $C(v)$.

### 3.4. The Manifold Log-ODE Method

Let $\mathcal{H}$ be a connected, cocommutative graded Hopf algebra, as detailed in Appendix A.1. By Theorem A.1, $\mathcal{H} \cong \mathcal{U}(\mathrm{Prim}(\mathcal{H}))$, so logarithmic signature increments can be expressed in primitive coordinates. For a path segment $X_k$, write its $\log_{\mathcal{H}}$-signature as $\lambda_k = \sum_{p \in \mathcal{B}_{\mathcal{H}}^{\mathrm{prim}}} \lambda_k^p p \in \mathrm{Prim}(\mathcal{H})$. Given the vector-field lift $F_W$, the corresponding log-ODE field is

$$L_k(Y) = \sum_{p \in \mathcal{B}_{\mathcal{H}}^{\mathrm{prim}}} \lambda_k^p F_W(p)(Y). \tag{8}$$

The lifted field $F_W(p)(Y)$ is evaluated by the recursions in Section 3.3: shuffle primitives use the Lie bracket recursion, while tree primitives use the multi-ary covariant derivative recursion. The vector-field lift is evaluated at the solver stages whenever $L_k$ is queried.

In Euclidean space, the segment update is obtained by solving the normalised ODE $\dot{Z}_\tau = L_k(Z_\tau)$, $Z_0 = Y_k$, $\tau \in [0, 1]$, where the segment length is already encoded in $\lambda_k$. We solve this using Heun's second-order method and recover log-NCDE when, additionally, $\mathcal{H} = \mathcal{H}_{\sqcup\sqcup}$.

While our method works for any connection-equipped manifold, we make the implementation choice of using homogeneous spaces. As such, we represent tangent vectors in frame or Lie-algebra coordinates. Let $G$ be the Lie group acting on $\mathcal{M}$, and write $g \cdot Y$ for the action. For $\xi \in \mathfrak{g}$, denote the induced fundamental vector field by

$$\xi^{\#}(Y) = \left.\frac{\mathrm{d}}{\mathrm{d}\epsilon}\right|_{\epsilon=0} \exp(\epsilon\xi) \cdot Y.$$

The lifted primitive evaluator returns frame coordinates $\widehat{F}_W(p)(Y) \in \mathfrak{g}$ satisfying

$$F_W(p)(Y) = \widehat{F}_W(p)(Y)^{\#}(Y).$$

Consequently, the window field is represented by

$$\widehat{L}_k(Y) = \sum_{p \in \mathcal{B}_{\mathcal{H}}^{\mathrm{prim}}} \lambda_k^p \widehat{F}_W(p)(Y), \quad L_k(Y) = \widehat{L}_k(Y)^{\#}(Y).$$

The log-ODE on the time interval $\tau \in [0, 1]$ is then

$$\dot{Z}_\tau = \widehat{L}_k(Z_\tau)^{\#}(Z_\tau), \qquad Z_0 = Y_k.$$

This is a geometric numerical integration problem that may be solved using Runge–Kutta–Munthe–Kaas or Commutator free (CF) methods. We select CF-EES$(2, 5)$ for its minimal exponential design which minimises per-step memory and compute costs (Shmelev et al., 2026). We refer the reader to their text and appendices for an overview of commutator-free methods in neural differential equations.

In the flat case, the group action reduces to addition, and the exponential update reduces to the corresponding explicit ODE method applied to (8). Hence, the same implementation covers the Euclidean $\mathcal{H}_{\mathrm{GL}}$ case as well as the homogeneous-space $\mathcal{H}_{\sqcup\sqcup}$ and $\mathcal{H}_{\mathrm{MKW}}$ cases.

### 3.5. Complexity Analysis

At truncation depth $N$, the algebraic cost is governed by the primitive basis size of the chosen Hopf algebra. Let $b_{\mathcal{H}}(n)$ denote the number of degree-$n$ primitive basis elements and let $d$ be the path dimension. For $\mathcal{H}_{\sqcup\sqcup}$, $b_{\sqcup\sqcup}(n)$ is given by Witt's formula (Witt, 1937); for $\mathcal{H}_{\mathrm{GL}}$, primitives are $d$-decorated non-planar rooted trees (Göbel, 1980); and for $\mathcal{H}_{\mathrm{MKW}}$, primitives are $d$-decorated planar rooted trees

([Walkup](), 1972). Hence

$$b_{\sqcup\sqcup}(n) = \frac{1}{n} \sum_{k|n} \mu(k)\, d^{n/k},$$

$$b_{\mathrm{GL}}(n) = t_n d^n,$$

$$b_{\mathrm{MKW}}(n) = C_{n-1} d^n = \frac{1}{n}\binom{2n-2}{n-1} d^n,$$

where $\mu$ is the Möbius function and $t_n$ is the number of non-planar rooted-tree shapes with $n$ vertices. Write

$$|\mathcal{B}_{\mathcal{H}}^{\mathrm{prim}}(N)| := \sum_{n=1}^{N} b_{\mathcal{H}}(n).$$

Assume all models use the same atomic vector field $f_\theta$, an $m$-layer multilayer perceptron (MLP) with hidden width $n_h$ and hidden-state dimension $u$. For $f_\theta \colon \mathbb{R}^u \to \mathbb{R}^{u\times d}$, one primal evaluation costs

$$C_\theta = 2un_h + 2(m-1)n_h^2 + 2udn_h.$$

Using the standard estimate that one JVP costs three primal evaluations, and reusing derivative evaluations across primitive fields, the Euclidean per-field costs are

$$\mathcal{C}_{\sqcup\sqcup}^{(N)} = 3\,|\mathcal{B}_{\sqcup\sqcup}^{\mathrm{prim}}(N-1)|\,C_\theta,$$

$$\mathcal{C}_{\mathrm{GL}}^{(N)} = 3\,|\mathcal{B}_{\mathrm{GL}}^{\mathrm{prim}}(N-1)|\,C_\theta.$$

For manifold-valued models, ordinary JVPs are replaced by covariant or coordinate directional derivatives. The $\mathcal{H}_{\sqcup\sqcup}$ lift evaluates each non-letter Lyndon primitive by a Lie bracket, which requires two directional derivative evaluations. Following the same JVP cost model, this gives

$$\mathcal{C}_{\sqcup\sqcup,\nabla}^{(N)} \approx 6\,|\mathcal{B}_{\sqcup\sqcup}^{\mathrm{prim}}(N-1)|\,C_\theta.$$

For $\mathcal{H}_{\mathrm{MKW}}$, a degree-$n$ planar tree has $n-1$ edges, and each edge corresponds to one covariant JVP in the ordered tree lift. Hence $D_{\mathrm{MKW}}(N) = \sum_{n=1}^{N}(n-1)b_{\mathrm{MKW}}(n)$, and

$$\mathcal{C}_{\mathrm{MKW},\nabla}^{(N)} \approx 3D_{\mathrm{MKW}}(N)C_\theta.$$

These estimates isolate the neural evaluation cost; manifold implementations also incur lower-order geometry costs from coordinate or frame conversion, connection operations, Lie-bracket structure terms, and group actions.

It remains to account for solver stages. With $J$ internal steps and $R$ field evaluations per step,

$$\mathcal{C}_{\mathcal{H},\mathrm{window}}^{(N)} = JR\,\mathcal{C}_{\mathcal{H}}^{(N)}, \quad \mathcal{H} \in \{\mathcal{H}_{\sqcup\sqcup}, \mathcal{H}_{\mathrm{GL}}\},$$

so Euclidean Heun gives $2J\,\mathcal{C}_{\mathcal{H}}^{(N)}$. For homogeneous-space models integrated by a commutator-free scheme with $R$ field-evaluation stages and $Q$ exponentials per step,

$$\mathcal{C}_{\mathcal{H},\mathrm{window}}^{(N)} \approx JR\,\mathcal{C}_{\mathcal{H},\nabla}^{(N)} + JQ\,C_{\mathcal{M}}, \quad \mathcal{H} \in \{\mathcal{H}_{\sqcup\sqcup}, \mathcal{H}_{\mathrm{MKW}}\}.$$

Here $C_{\mathcal{M}}$ denotes the cost of the group exponential and action. For CF-EES$(2,5)$, $Q = R = 3$; in our implementation, each exponential uses a fourth-order approximation requiring two matrix multiplications ([Sastre et al.](), 2019).

*Table 2.* Per-window costs for $N = 2$. Here $P_2$ is the primitive dimension up to depth 2, $A_2$ is the neural derivative multiplier so that one field evaluation costs $3A_2C_\theta$, and $(R, Q)$ denotes field-evaluation stages and exponentials per solver step. $\mathbb{R}, \mathcal{H}_{\sqcup\sqcup}$ represents log-NCDE.

| Space/Algebra | $P_2$ | $A_2$ | $(R,Q)$ | Cost/window |
|---|---|---|---|---|
| $\mathbb{R}, \mathcal{H}_{\sqcup\sqcup}$ | $\frac{d(d+1)}{2}$ | $d$ | $(2,0)$ | $6JdC_\theta$ |
| $\mathbb{R}, \mathcal{H}_{\mathrm{GL}}$ | $d + d^2$ | $d$ | $(2,0)$ | $6JdC_\theta$ |
| $\mathcal{M}, \mathcal{H}_{\sqcup\sqcup}$ | $\frac{d(d+1)}{2}$ | $2d$ | $(3,3)$ | $18JdC_\theta + 3JC_{\mathcal{M}}$ |
| $\mathcal{M}, \mathcal{H}_{\mathrm{MKW}}$ | $d + d^2$ | $d^2$ | $(3,3)$ | $9Jd^2C_\theta + 3JC_{\mathcal{M}}$ |

### 3.6. The Branched Signature Kernel

The geometric signature kernel between two paths $x, y$ is

$$k_{\mathrm{geo}}(x,y) = \langle \mathrm{Sig}^N(x), \mathrm{Sig}^N(y)\rangle,$$

where the inner product is taken on the depth-$N$ truncated tensor signature space. Signature kernels provide a principled similarity measure for stochastic processes ([Chevyrev & Oberhauser](), 2022) and have been used to train neural SDEs through signature-kernel scoring objectives ([Issa et al.](), 2023). In practice, training minimises

$$\begin{aligned} \mathcal{L}_{\mathrm{geo}}(\theta) = &\,\mathbb{E}_{X,X'\sim P_\theta}\big[k_{\mathrm{geo}}(X,X')\big] \\ &- 2\mathbb{E}_{X\sim P_\theta, Y\sim P_{\mathrm{data}}}\big[k_{\mathrm{geo}}(X,Y)\big], \end{aligned} \quad (9)$$

where $P_{\mathrm{data}}$ denotes the data law and $P_\theta$ the model law. The term $\mathbb{E}_{Y,Y'\sim P_{\mathrm{data}}}\big[k_{\mathrm{geo}}(Y,Y')\big]$ is constant in $\theta$.

Existing scalable signature-kernel constructions ([Salvi et al.](), 2021; [Cass et al.](), 2025; [Tóth](), 2025; [Tóth et al.](), 2025) are based on geometric signatures, and hence compare paths through coordinates corresponding to iterated Stratonovich integrals. For semimartingale data observed on a grid, the canonical piecewise-linear geometric lift lies in the Wong–Zakai regime and does not expose quadratic covariation as an explicit driver coordinate. Consequently, an Itô model using this representation must recover bracket information indirectly from the grid via squared increments.

**Definition 3.1** (Branched signature kernel objective)**.** Fix a Hopf algebra $\mathcal{H}$, truncation depth $N$, and the coordinate basis used by $\mathrm{Sig}_{\mathcal{H}}^N$; throughout, $\langle \cdot, \cdot \rangle_{\mathcal{H}_{\leq N}}$ denotes the Euclidean inner product on these coordinates. For drivers $\mathbf{X}, \mathbf{Y}$ enhanced with their quadratic variation, define

$$k_{\mathrm{br}}^N(\mathbf{X}, \mathbf{Y}) = \big\langle \mathrm{Sig}_{\mathcal{H}}^N(\mathbf{X}), \mathrm{Sig}_{\mathcal{H}}^N(\mathbf{Y})\big\rangle_{\mathcal{H}_{\leq N}}.$$

The branched signature-kernel objective is

$$\begin{aligned} \mathcal{L}_{\mathrm{br}}(\theta) = &\,\mathbb{E}_{\mathbf{X},\mathbf{X}'\sim P_\theta}\big[k_{\mathrm{br}}^N(\mathbf{X},\mathbf{X}')\big] \\ &- 2\mathbb{E}_{\mathbf{X}\sim P_\theta, \mathbf{Y}\sim P_{\mathrm{data}}}\big[k_{\mathrm{br}}^N(\mathbf{X},\mathbf{Y})\big]. \end{aligned} \quad (10)$$

As in (9), the omitted data–data term is constant in $\theta$.

When bracket increments are available from the simulator, the objective uses them directly rather than reconstructing them from realised quadratic variation. Appendix B.2 formalises this finite-grid distinction. A geometric kernel computed from path values is a kernel on the projected path law, whereas the branched kernel is computed on the enhanced law containing bracket coordinates. A hypothetical bracket-aware baseline built only from grid samples must first reconstruct those coordinates, and is therefore subject to realised-covariance error. For standard continuous semimartingales, fixed-time realised-covariance fluctuations are typically of order $\sqrt{\Delta_n}$, so supplying analytic or simulator-ground-truth brackets can remove this finite-grid error source without changing the continuous-limit law-matching target.

## 4. Experiments

We design our experimental evaluation to focus on regimes where standard Euclidean NRDEs struggle. We validate B-NRDE across three distinct domains, mapping each to the Hopf algebra that captures its geometry and causality.

We first consider Euclidean rough volatility. Utilising $\mathcal{H}_{\mathrm{GL}}$, we task B-NRDE with learning an unconditional generative model of the rough Bergomi stochastic volatility model, evaluating performance via the fidelity of the marginal laws. Next, we examine Stratonovich manifold dynamics under $\mathcal{H}_{\sqcup\!\sqcup}$ and evaluate forecasting performance. Finally, we address manifold-valued Itô rough paths, employing $\mathcal{H}_{\mathrm{MKW}}$ to learn a generative model of covariance dynamics on the symmetric positive definite (SPD) matrix manifold. Dataset code is available in RoughBench, and further experimental details may be found in Appendix B.

### 4.1. Baseline Models

As baselines, we employ manifold neural ODE (M-NODE) (Lou et al., 2020), NCDE with linear (Kidger et al., 2020), Hermite (Morrill et al., 2022), and Savitzky–Golay (SG) (Bastian et al., 2025) interpolation, dubbed NCDE, NCDE++ and SG-NCDE, respectively. We also consider signature methods NRDE (Morrill et al., 2021b) and log-NCDE (Walker et al., 2024), and discrete-time GRU (Chung et al., 2014), xLSTM (Beck et al., 2024), and stacked xLSTM, the backbone of Beck et al. (2024).

### 4.2. Generative Modeling of Rough Volatility Under Itô Integration

Rough volatility models posit that log-volatility evolves with fractional regularity ($H < \frac{1}{2}$) (Gatheral et al., 2014). A prominent example is the rough Bergomi (rBergomi) model (Bayer et al., 2015), where volatility is driven by a

*Table 3.* rBergomi KS across time-marginals

| Method | Training Time (s) | KS Score ($\times 10^{-2}$) | | | |
|---|---|---|---|---|---|
| | | 128 | 256 | 384 | 512 |
| xLSTM | 38 | $11.04_{\pm 1.38}$ | $10.31_{\pm 1.30}$ | $10.94_{\pm 0.89}$ | $11.71_{\pm 0.71}$ |
| NCDE | 1154 | $9.06_{\pm 1.94}$ | $8.93_{\pm 1.07}$ | $9.75_{\pm 1.02}$ | $9.65_{\pm 0.81}$ |
| NCDE++ | 12021 | $8.11_{\pm 0.93}$ | $8.82_{\pm 0.86}$ | $9.67_{\pm 0.34}$ | $10.73_{\pm 0.52}$ |
| Stacked xLSTM | 313 | $\underline{7.69}_{\pm 0.16}$ | $9.19_{\pm 1.09}$ | $9.50_{\pm 1.21}$ | $10.95_{\pm 1.11}$ |
| log-NCDE | 74 | $7.75_{\pm 1.39}$ | $\underline{7.38}_{\pm 0.46}$ | $8.72_{\pm 0.88}$ | $8.89_{\pm 0.24}$ |
| NRDE | 157 | $8.74_{\pm 0.96}$ | $8.41_{\pm 0.94}$ | $8.12_{\pm 0.79}$ | $\mathbf{7.47}_{\pm \mathbf{0.83}}$ |
| GRU | 42 | $7.73_{\pm 1.78}$ | $7.59_{\pm 0.58}$ | $\underline{7.87}_{\pm 0.67}$ | $\underline{8.02}_{\pm 1.15}$ |
| B-NRDE (GK) | 233 | $\mathbf{6.89}_{\pm \mathbf{1.81}}$ | $\mathbf{6.91}_{\pm \mathbf{0.98}}$ | $8.56_{\pm 0.38}$ | $9.58_{\pm 0.36}$ |
| **B-NRDE (BK)** | 47 | $7.93_{\pm 1.49}$ | $7.70_{\pm 0.87}$ | $\mathbf{7.77}_{\pm \mathbf{1.02}}$ | $8.11_{\pm 0.52}$ |

Volterra process $W_t^H = \int_0^t K_H(t - s)\mathrm{d}W_s$. Simulating these paths is a known bottleneck, as exact methods scale quadratically with the number of time steps. Furthermore, standard signature-based approaches are constrained to the geometric (Stratonovich) setting; capturing the financially-relevant Itô integral requires augmenting the path with a lead-lag (Hoff) lift (Flint et al., 2016), doubling channel dimension from $d \to 2d$, massively growing signature size.

We use rBergomi to stress-test the Itô capability of the B-NRDE framework in flat geometry ($\mathcal{M} = \mathbb{R}^d$). Unlike NRDE, our $\mathcal{H}_{\mathrm{GL}}$ formulation accommodates branched rough paths, allowing B-NRDE to process the driving signal without lead–lag augmentation. Equation and simulation details can be found in Appendix D.1. For training, we generate a time-augmented latent piecewise-linear Brownian motion $(t, Z_{\mathrm{lat}})$ on $[0, 1]$ with grid size $\Delta_n$, and sample generated log-price paths $X := \widehat{\log S}_\theta \sim P_\theta$. Ground-truth paths are $Y := \log S \sim P_{\mathrm{data}}$. All models minimise $\mathcal{L}_{\mathrm{geo}}$ except B-NRDE, which receives three additional fine-tuning epochs under the branched signature-kernel score between $(t, X)$ and $(t, Y)$, adding quadratic-variation/covariation terms $\langle X^i, X^j \rangle$ and $\langle Y^i, Y^j \rangle$; the former are the squared increments, the latter from the simulator ground truth.

In Table 3 we report Kolmogorov-Smirnov (KS) distances between model-generated and ground truth time-marginal laws. B-NRDE achieves the strongest fit across three of four horizons, outperforming $\mathcal{H}_{\sqcup\!\sqcup}$-based NRDE and log-NCDE.

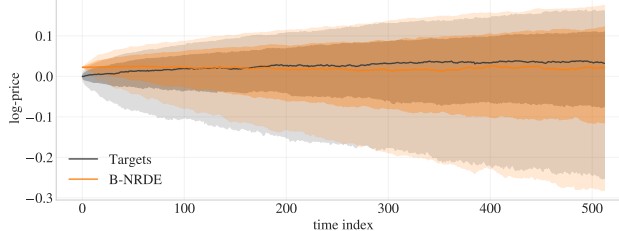

*Figure 3.* rBergomi sample paths generated by B-NRDE showing good fit to the underlying law.

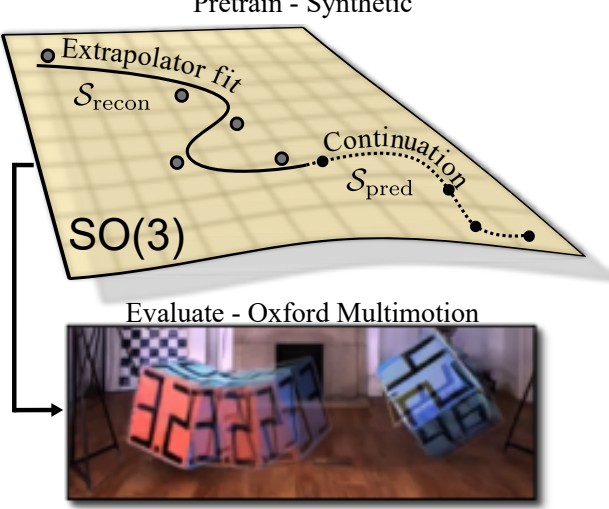

Pretrain - Synthetic

Evaluate - Oxford Multimotion

*Figure 4.* Schematic description of the sim-to-real dynamics training pipeline. Adapted from Bastian et al. (2025).

### 4.3. Sim-to-Real Dynamics Forecasting

Three-dimensional motion forecasting is central to robotics and computer vision, with applications to pose prediction (Yang et al., 2021; Dünkel et al., 2024) and occlusion-robust perception (Di et al., 2021). We consider sim-to-real forecasting of rotational dynamics, previously out of reach for signature methods due to the lack of a manifold-compatible formulation. As a deterministic task without an Itô dynamic, we leverage a coordinate-free $\mathcal{H}_{\sqcup\sqcup}$ formulation.

Following Bastian et al. (2025), we generate offline continuous-time rotation trajectories $R \colon [0,1] \to \mathrm{SO}(3)$, where $R(t)$ is the orientation at time $t$. From each trajectory we extract sliding windows $\{(t_j, R(t_j))\}_{j=0}^m$ and split each window into a reconstruction prefix $\mathcal{S}_{\mathrm{recon}}$ and prediction suffix $\mathcal{S}_{\mathrm{pred}}$. An extrapolator E is fit on $\mathcal{S}_{\mathrm{recon}}$ and evaluated on the full window to produce a dense input path $\tilde{R}(t)$ over $\mathcal{S}_{\mathrm{recon}} \cup \mathcal{S}_{\mathrm{pred}}$. This extrapolated path is then provided to the model, which outputs predicted rotations $\{\hat{R}_\theta(t_j)\}_{j=0}^m$ over the same times. We pretrain by minimising the Frobenius discrepancy to the ground truth over the full window:

$$\min_\theta \ \sum_{j \in \mathcal{J}} \left\| \hat{R}_\theta(t_j) - R(t_j) \right\|_F^2, \quad \hat{R}_\theta(t_j), \ R(t_j) \in \mathrm{SO}(3),$$

where $\mathcal{J} = \{0, \ldots, m\}$.

To assess sim-to-real generalisation, simulation-pretrained models are evaluated on the Oxford Multi-motion Dataset (OMD) (Judd & Gammell, 2019), with windows and extrapolations identical to those used for the simulation data. The rotation geodesic error (RGE) (Huynh, 2009), defined as $\mathrm{RGE}(R_1, R_2) = 2\arcsin\left(\frac{\|R_2 - R_1\|_F}{2\sqrt{2}}\right)$, is computed over both segments. Figure 4 illustrates the protocol.

*Table 4.* Training test set Frobenius norm and RGE (degrees) across different motion scenarios in the OMD dataset.

| Method | Pretraining Frobenius | Static Motion | Translation Motion | Unconstrained Motion |
|---|---|---|---|---|
| M-NODE | 2.461 | 132.40 | 118.25 | 121.08 |
| SO(3)-NCDE | 0.294 | 22.21 | 21.83 | 22.45 |
| SO(3)-GRU | 0.271 | 20.25 | 20.39 | 21.13 |
| xLSTM | 0.233 | 16.67 | 16.77 | 17.36 |
| NRDE | 0.097 | 7.45 | 7.34 | 7.32 |
| Stacked xLSTM | 0.110 | 7.11 | 7.50 | 7.03 |
| SG-NCDE | 0.050 | **2.93** | **3.31** | **2.80** |
| **B-NRDE (GK)** | **0.049** | 3.23 | 3.70 | 3.33 |

Whilst SG-NCDE attains the lowest RGE, B-NRDE achieves comparable accuracy with only two solver steps versus 20. The remaining small deficit is not explained by the simulation pretraining loss, suggesting modest degradation in sim-to-real transfer. Crucially, B-NRDE generalises the SG-NCDE pipeline by admitting rough drivers, thereby removing the $C^1$ interpolant requirement and permitting non-smooth extrapolators such as MLPs.

### 4.4. Itô Dynamics over the SPD Manifold

SPD-covariance dynamics learning arises in multivariate finance (Noureldin et al., 2012; Johansson et al., 2023), medical imaging (Pennec, 2020), and, more recently, Riemannian diffusion models (Park et al., 2022; De Bortoli et al., 2022; Thornton et al., 2022). Notably, finance canonically requires non-anticipative Itô modelling to avoid lookahead bias, and diffusion models are typically formulated as Itô diffusions.

As such, we study a mean-reverting Itô diffusion on the SPD manifold $\mathbb{S}_d^{++}$ endowed with the affine invariant (AI) geometry, testing B-NRDE's ability to learn non-anticipative stochastic dynamics on a curved state space while strictly preserving the SPD cone. Let $M \in \mathbb{S}_d^{++}$ and let $X_0 \in \mathbb{S}_d^{++}$. Consider the OU-type Itô diffusion

$$\mathrm{d}X_t = \eta \log_{X_t}(M)\, \mathrm{d}t + \sigma\, X_t^{1/2}\, \mathrm{d}B_t\, X_t^{1/2}, \quad (11)$$

where $\eta > 0$ controls mean reversion, $\sigma \geq 0$ is the noise scale, and $B_t \in \mathrm{Sym}(d)$ is a symmetric matrix Brownian motion. In practice, the vectorised state $\tilde{x}_t = \mathrm{vech}(X_t) \in \mathbb{R}^q$, $\mathrm{d}\langle \tilde{x} \rangle_t \in \mathbb{R}^{q \times q}$ with $q = \frac{d(d+1)}{2}$ is used throughout.

Training minimises an expected signature-kernel objective between the ground-truth and generated trajectories, with log-NCDE using the geometric signature kernel and B-NRDE. Since $\mathbb{S}_d^{++}$ is not totally ordered, a KS statistic is not directly applicable. We therefore sort the eigenvalues increasingly and report empirical time-marginal $W_1$ distances between the eigenvalue vectors $\lambda(X_t), \lambda(\hat{X}_t) \in \mathbb{R}^d$.

In Table 5 we find that the manifold-preserving capabilities of B-NRDE improve on the results of Euclidean log-NCDE

in terms of law-matching, showing an average improvement of 56.2%. However, visual inspection of Figure 5 shows somewhat disappointing fidelity near the initial condition, a common pitfall of signature-based methods that are invariant to the starting value without augmentation (Morrill et al., 2021a). We also find no meaningful performance difference between the branched and geometric kernels in this setting.

*Table 5.* 1-Wasserstein distance between predicted and ground-truth eigenvalue marginals on $\mathbb{S}_d^{++}$.

| Method | Training Time (s) | 1-Wasserstein ($\times 10^{-2}$) | | | |
|---|---|---|---|---|---|
| | | **128** | **256** | **384** | **512** |
| NCDE++ | 1605 | $64.95_{\pm 0.98}$ | $58.27_{\pm 1.36}$ | $50.56_{\pm 1.73}$ | $42.15_{\pm 2.22}$ |
| NCDE | 1241 | $65.56_{\pm 1.74}$ | $58.72_{\pm 2.38}$ | $50.32_{\pm 3.02}$ | $42.38_{\pm 3.11}$ |
| NRDE | 232 | $11.80_{\pm 0.15}$ | $15.50_{\pm 0.16}$ | $17.36_{\pm 0.13}$ | $17.70_{\pm 0.28}$ |
| xLSTM | 12 | $8.36_{\pm 0.63}$ | $10.49_{\pm 0.41}$ | $13.16_{\pm 0.39}$ | $14.09_{\pm 0.53}$ |
| log-NCDE | 481 | $12.10_{\pm 0.30}$ | $14.76_{\pm 0.22}$ | $16.17_{\pm 0.32}$ | $16.81_{\pm 0.40}$ |
| Stacked xLSTM | 18 | $\mathbf{5.50_{\pm 0.10}}$ | $6.85_{\pm 0.35}$ | $11.47_{\pm 0.63}$ | $14.30_{\pm 0.46}$ |
| **B-NRDE (GK)** | 495 | $\underline{5.73_{\pm 0.20}}$ | $\mathbf{5.81_{\pm 0.19}}$ | $\mathbf{6.28_{\pm 0.86}}$ | $\underline{8.35_{\pm 0.94}}$ |
| **B-NRDE (BK)** | 502 | $5.81_{\pm 0.20}$ | $\underline{5.87_{\pm 0.21}}$ | $\underline{6.30_{\pm 0.85}}$ | $\mathbf{8.32_{\pm 0.94}}$ |

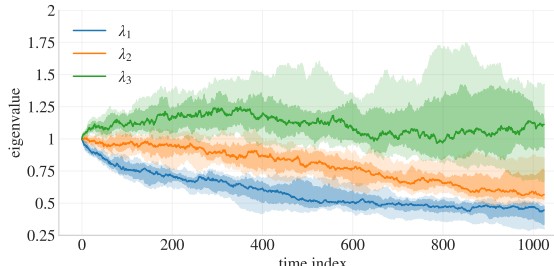

*(a)* Ground truth eigenvalue trajectories of Equation (11).

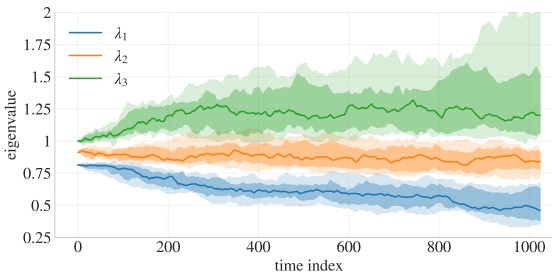

*(b)* Predicted eigenvalue trajectories output by B-NRDE.

*Figure 5.* Ground truth vs. B-NRDE eigenvalue trajectories.

# 5. Discussion

In this work, we introduced B-NRDE, a generalisation of NRDE to branched rough paths, and a branched-signature-kernel training objective for neural stochastic differential equations (SDEs). Our method extends signature methods beyond the geometric/Stratonovich setting to Itô, manifold, and Itô-on-manifold dynamics, while remaining empirically competitive. We summarise key limitations of the current method and outline the most promising extensions.

## 5.1. Limitations

Unlike the untruncated PDE-based solvers available for geometric kernels (Salvi et al., 2021), branched signature kernels require explicit truncation, discarding higher-order iterated-integral information at greater memory and compute cost. We expect this to worsen as the state dimension $d$ grows, since the quadratic-variation features scale as $d^2$.

## 5.2. Future Directions

**Numerical** Such PDE-based solvers, or other approximation schemes, offer the most promising route to avoiding explicit truncation in our kernel. Adaptive log-ODE schemes could also permit larger signature windows and faster integration (Bayer et al., 2023).

**Signatures** The log-signature quotients out the shuffle identities, projecting onto the strictly smaller Lyndon-word basis. No analogous projection is known explicitly for the branched Hopf algebras used here, though recent work (Bellingeri et al., 2024) suggests one exists; this would yield smaller branched log-signatures and improve the memory efficiency of B-NRDE. For scalar RDEs, such as the rough Bergomi model of Section 4.2, the theory of multi-index rough paths, elements of the Linares–Otto-Tempelmayr Hopf algebra (Linares et al., 2023), offers an even more compact representation by exploiting symmetries in the vector fields of scalar equations (Bellingeri et al., 2026).

**SPDEs** The rough-path philosophy extends from finite-dimensional differential equations to stochastic partial differential equations (SPDEs) via the theory of regularity structures (Hairer, 2014). Much like how planarly branched rough paths enable solving RDEs over manifolds, planar regularity structures (Rahm, 2022) provide the corresponding generalisation for regularity structures and SPDEs. Figure 6 recapitulates Rahm (2022, Fig. 1) and points to a natural future direction: a neural planar-regularity-structure model in a $\mathbb{R}^d \to \mathcal{M}$ setting, beyond the $\mathbb{R} \to \mathcal{M}$ framework of B-NRDE and the $\mathbb{R}^d \to \mathbb{R}^d$ framework of Neural Operator with Regularity Structure (Hu et al., 2022) and Deep Latent Regularity Net (Gong et al., 2023).

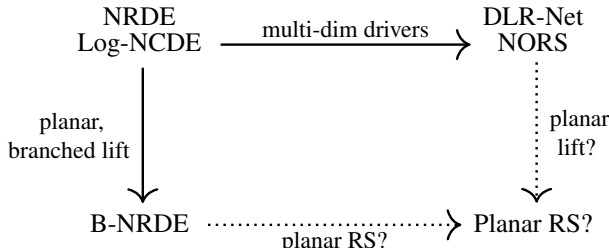

*Figure 6.* Positioning of our method and a hypothetical planar-regularity-structure extension.

## Impact Statement

This paper presents fundamental work in time-series forecasting whose goal is to advance the field of Machine Learning. There are many potential societal consequences of our work, but none of which we feel must be specifically highlighted here.

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

# A. The Hopf Algebras of Rough Paths

This appendix records the algebraic structures used throughout the paper. We first detail the formal requirements imposed on our Hopf algebras, then explain why we use (i) the *graded dual* of the shuffle Hopf algebra for geometric, Stratonovich-type drivers, (ii) Grossman–Larson (GL) rather than Butcher–Connes–Kreimer (BCK) for branched, Itô-type drivers, and (iii) the *graded dual* of the Munthe–Kaas–Wright (MKW) Hopf algebra for the manifold log-ODE scheme. We keep the presentation concise and refer to Kern & Lyons (2023) and Manchon (2025) for a more comprehensive account.

## A.1. Hopf Algebras

**Gradedness.** A Hopf algebra $\mathcal{H}$ is graded if it admits a decomposition

$$\mathcal{H} = \bigoplus_{n \geq 0} \mathcal{H}_n$$

such that its product $\star$ and coproduct $\Delta$ respect degree:

$$\mathcal{H}_m \star \mathcal{H}_n \subseteq \mathcal{H}_{m+n}, \qquad \Delta(\mathcal{H}_n) \subseteq \bigoplus_{m=0}^{n} \mathcal{H}_m \otimes \mathcal{H}_{n-m}. \tag{12}$$

The component $\mathcal{H}_n$ corresponds to level $n$ signature coordinates. For word signatures, $n$ is word length; for branched signatures, $n$ is the number of vertices.

**Connectedness.** A graded Hopf algebra is connected if the degree-zero component is spanned by the unit:

$$\mathcal{H}_0 = \mathrm{span}\{1\}.$$

For instance, in the GL Hopf algebra on rooted forests, the empty forest is the unit $1$, and every non-empty forest has positive degree.

**Truncation.** In applications, we truncate at depth $N$:

$$\mathcal{H}^{(\leq N)} := \bigoplus_{n=0}^{N} \mathcal{H}_n.$$

This truncation matches the depth of the signature or log-signature retained in the numerical scheme.

**Primitives and grouplike elements.** Primitive elements have no nontrivial coproduct:

$$\mathrm{Prim}(\mathcal{H}) := \{p \in \mathcal{H} : \Delta p = p \otimes 1 + 1 \otimes p\}.$$

For rough paths, increments are characters of the coordinate Hopf algebra; equivalently, they are grouplike elements of the graded dual. The Hopf logarithm $\log_\star$, as in Equation (6), maps grouplike elements to primitive elements.

## A.2. Why cocommutativity matters: Milnor–Moore

The log-ODE construction is expressed at the level of primitives: one takes the Hopf logarithm of a grouplike increment, obtains a Lie element, and then exponentiates the resulting truncated Lie polynomial in differential operators. This gives a compression from the full Hopf algebra to its primitive part. To recover the Hopf algebra from its primitives, we use the Milnor–Moore theorem.

**Theorem A.1** (Milnor–Moore (Milnor & Moore, 1965)). *Let $\mathcal{H}$ be a connected, graded, cocommutative Hopf algebra over a field of characteristic $0$. Then there is a canonical Hopf algebra isomorphism*

$$\mathcal{H} \cong U\big(\mathrm{Prim}(\mathcal{H})\big). \tag{13}$$

Consequently, we do not use the noncocommutative coordinate Hopf algebras BCK and MKW directly for the primitive log-ODE construction. Instead, we use their cocommutative graded duals, namely $GL$ and $(MKW)^\circ$.

### A.3. Words and Stratonovich calculus

A.3.1. WORDS AND LYNDON WORDS

Throughout this subsection, let $A$ be a finite alphabet, typically $A = \{1, \ldots, d\}$, with a fixed total order. A *word* is a finite sequence $w = i_1 \cdots i_n$ with $i_k \in A$, including the empty word $1$ when $n = 0$. Its length is $|w| = n$, concatenation is denoted by juxtaposition, and the order on $A$ induces the lexicographic order on words.

**Definition A.2** (Lyndon word). A nonempty word $\ell$ is a *Lyndon word* if it is strictly smaller, in lexicographic order, than each of its nontrivial proper suffixes.

**Example A.3** (Lyndon bracketing). *Let $A = \{a, b, c, d\}$ with $a < b < c < d$. The word $\ell = \texttt{abac}$ is Lyndon, since*

$$\texttt{abac} < \texttt{bac}, \qquad \texttt{abac} < \texttt{ac}, \qquad \texttt{abac} < \texttt{c}.$$

*Its standard factorisation is $\ell = uv$, with $u = \texttt{ab}$ and $v = \texttt{ac}$. Hence*

$$[\ell] = [[u], [v]] = [[e_a, e_b], [e_a, e_c]].$$

*As $\ell$ ranges over Lyndon words, these bracketings form the Lyndon basis of the free Lie algebra $\mathrm{Prim}(\mathcal{H}_{\shuffle})$.*

A.3.2. STRATONOVICH ROUGH PATHS ON $\mathbb{R}^n$: THE TENSOR HOPF ALGEBRA

We view the geometric signature as a grouplike element of the cocommutative graded dual of the shuffle Hopf algebra,

$$\mathcal{H}_{\shuffle}^\circ := \big(T(A), \star_{\shuffle}, \Delta_{\mathrm{unsh}}\big),$$

where $\star_{\shuffle}$ is concatenation and $\Delta_{\mathrm{unsh}}$ is the unshuffle coproduct. This is the graded dual of the usual shuffle Hopf algebra $(T(A), \shuffle, \Delta_{\mathrm{dec}})$. For words $u = i_1 \cdots i_m$ and $v = j_1 \cdots j_n$,

$$u \star_{\shuffle} v := i_1 \cdots i_m j_1 \cdots j_n.$$

The unshuffle coproduct is

$$\Delta_{\mathrm{unsh}}(i_1 \cdots i_n) := \sum_{I \subseteq [n]} i_I \otimes i_{I^c}, \qquad i_I := i_{k_1} \cdots i_{k_{|I|}} \quad \text{for } I = \{k_1 < \cdots < k_{|I|}\}.$$

It is cocommutative, since swapping tensor factors corresponds to $I \leftrightarrow I^c$. The primitives $\mathrm{Prim}(\mathcal{H}_{\shuffle}^\circ)$ form the free Lie algebra on $A$, with bracket induced by the commutator of $\star_{\shuffle}$. Therefore the Hopf logarithm of any geometric signature increment is a Lie element, and may be expanded in the Lyndon basis.

**Example A.4** (Smooth tensor signature). *Let $X_t^a = t$ and $X_t^b = t^2$ on $[0, 1]$. Then*

$$\langle S(X), a \rangle = \int_0^1 dX_t^a = 1, \qquad \langle S(X), b \rangle = \int_0^1 dX_t^b = 1,$$

*and*

$$\langle S(X), ab \rangle = \int_{0 < u < v < 1} dX_u^a \, dX_v^b = \int_0^1 v \, 2v \, dv = \frac{2}{3},$$

*while*

$$\langle S(X), ba \rangle = \int_{0 < u < v < 1} dX_u^b \, dX_v^a = \int_0^1 v^2 \, dv = \frac{1}{3}.$$

*Thus*

$$\langle S(X), ab + ba \rangle = \langle S(X), a \rangle \langle S(X), b \rangle,$$

*as required by the shuffle character relation. With $V_a = \partial_y$ and $V_b = y \partial_y$, and with the convention that $ab$ acts by $V_a V_b$,*

$$V_a V_b \varphi = \varphi' + y \varphi'', \qquad V_b V_a \varphi = y \varphi'',$$

*so the two tensor coordinates act on distinct differential operators even for this one-dimensional example.*

## A.4. Rooted trees and Itô calculus

### A.4.1. NON-PLANAR AND PLANAR ROOTED TREES

**Definition A.5** (Rooted tree). A rooted tree is a finite directed acyclic graph with a distinguished root and optional decoration by elements of an alphabet $A$. The tree with root decorated by $i$ and children $\tau_1, \ldots, \tau_n$ is denoted

$$[\tau_1, \ldots, \tau_n]_i = \vcenter{\hbox{\includegraphics{}}}^{\tau_1 \cdots \tau_n}_i.$$

The order $|\tau|$ is the number of vertices, and a forest is a finite product of rooted trees.

We distinguish *non-planar* rooted trees, whose children are unordered, from *planar* rooted trees, whose children are ordered from left to right. Thus, in the planar setting,

$$\vcenter{\hbox{}}^{j \quad k}_i \neq \vcenter{\hbox{}}^{k \quad j}_i,$$

whereas these two trees agree in the non-planar setting.

### A.4.2. PRE-LIE GRAFTING AND EUCLIDEAN ELEMENTARY DIFFERENTIALS

The Euclidean branched theory is governed by a pre-Lie product. A pre-Lie algebra is a vector space with a bilinear product $\triangleright$ satisfying

$$x \triangleright (y \triangleright z) - (x \triangleright y) \triangleright z = y \triangleright (x \triangleright z) - (y \triangleright x) \triangleright z.$$

For vector fields on $\mathbb{R}^n$, this product is

$$U \triangleright V := DV[U].$$

The free pre-Lie algebra is represented by non-planar rooted trees, with product given by grafting. For $\tau, \sigma \in \mathcal{UT}$,

$$\tau \curvearrowright \sigma := \sum_{v \in V(\sigma)} \tau \curvearrowright_v \sigma,$$

where $\tau \curvearrowright_v \sigma$ attaches the root of $\tau$ to the vertex $v$ of $\sigma$. The Euclidean elementary differential map satisfies

$$F(\bullet_i) = V_i, \qquad F([\tau_1, \ldots, \tau_k]_i) = D^k V_i\big(F(\tau_1), \ldots, F(\tau_k)\big),$$

and the grafting identity

$$F(\tau \curvearrowright \sigma) = F(\tau) \triangleright F(\sigma).$$

### A.4.3. BRANCHED ROUGH PATHS IN $\mathbb{R}^n$: GL AS THE COCOMMUTATIVE DUAL OF BCK

Itô integration introduces quadratic variation corrections, and word-indexed shuffle coordinates do not close under the Itô chain rule. Branched rough paths restore closure by indexing the expansion by rooted trees (Hairer & Kelly, 2014). The coordinate Hopf algebra is traditionally the noncocommutative BCK Hopf algebra, while the primitive log-ODE construction uses its cocommutative graded dual, the GL Hopf algebra (Grossman & Larson, 1989).

The GL product is the Guin–Oudom product extending pre-Lie grafting. If $\Delta_{\sqcup\sqcup}\alpha = \sum \alpha_{(1)} \otimes \alpha_{(2)}$ denotes the cocommutative unshuffle coproduct on forests, then

$$\alpha \star_{\mathrm{GL}} \beta = \sum \alpha_{(1)}\big(\alpha_{(2)} \curvearrowright \beta\big),$$

with the standard extension of $\curvearrowright$ from trees to forests (Oudom & Guin, 2004). In particular, for trees $\tau, \sigma \in \mathcal{UT}$,

$$\tau \star_{\mathrm{GL}} \sigma = \tau\sigma + \tau \curvearrowright \sigma.$$

For example,

$$\bullet_m \curvearrowright \vcenter{\hbox{}}^{j}_i = \vcenter{\hbox{}}^{m}_{i,j} + \vcenter{\hbox{}}^{j \quad m}_i.$$

The Hopf algebra

$$\mathcal{H}_{\mathrm{GL}} = \big(S(\mathcal{UT}), \star_{\mathrm{GL}}, \Delta_{\sqcup\sqcup}\big)$$

is connected, graded, and cocommutative. Hence Theorem A.1 identifies it with the enveloping Hopf algebra of its primitive Lie algebra.

**Example A.6** (Smooth branched signature). *Let $X_t^a = t$ and $X_t^b = t^2$ on $[0, 1]$. The canonical smooth branched lift is defined recursively by*

$$\langle \mathbf{X}_{0,t}, \bullet_i \rangle = X_t^i - X_0^i, \qquad \langle \mathbf{X}_{0,t}, [\tau_1, \ldots, \tau_k]_i \rangle = \int_0^t \prod_{r=1}^k \langle \mathbf{X}_{0,s}, \tau_r \rangle \, dX_s^i.$$

*Thus*

$$\langle \mathbf{X}, {}^{\bullet a}_{\bullet b} \rangle = \int_0^1 s \, d(s^2) = \frac{2}{3}, \qquad \langle \mathbf{X}, {}^{\bullet b}_{\bullet a} \rangle = \int_0^1 s^2 \, ds = \frac{1}{3},$$

*and, since the non-planar forest product is commutative,*

$$\langle \mathbf{X}, {}^{\bullet a} \diagdown\!\!\!\!\diagup {}^{\bullet b}_{\phantom{b} b} \rangle = \int_0^1 s \, s^2 \, d(s^2) = \frac{2}{5}.$$

*With $V_a = \partial_y$ and $V_b = y^2 \partial_y$ on $\mathbb{R}$,*

$$F\left({}^{\bullet a}_{\bullet b}\right) = DV_b[V_a] = 2y\partial_y, \qquad F\left({}^{\bullet a}\diagdown\!\!\!\!\diagup{}^{\bullet b}_{\phantom{b} b}\right) = D^2 V_b[V_a, V_b] = 2y^2 \partial_y.$$

*The same rooted-tree coordinates therefore encode both the Itô-type branched signature and the Euclidean elementary differentials appearing in the expansion.*

### A.4.4. POST-LIE GRAFTING AND MANIFOLD ELEMENTARY DIFFERENTIALS

On manifolds, the composition of differential operators is not represented by the Euclidean pre-Lie product alone. For a smooth connection $\nabla$, the MKW elementary differentials used below are defined through total covariant derivatives. When $\nabla$ has zero curvature and parallel torsion $T$, this construction admits the post-Lie interpretation

$$U \triangleright V := \nabla_U V, \qquad [U, V]_\nabla := -T(U, V).$$

The induced Lie bracket is

$$[\![U, V]\!] = U \triangleright V - V \triangleright U + [U, V]_\nabla,$$

which agrees with the Jacobi bracket of vector fields. Under the same flat/parallel-torsion assumptions, the post-Lie identities are

$$x \triangleright [y, z]_\nabla = [x \triangleright y, z]_\nabla + [y, x \triangleright z]_\nabla,$$

and

$$[x, y]_\nabla \triangleright z = x \triangleright (y \triangleright z) - (x \triangleright y) \triangleright z - y \triangleright (x \triangleright z) + (y \triangleright x) \triangleright z.$$

Planar rooted trees give the free combinatorial model of the resulting post-Lie calculus. The relevant grafting operation is left grafting:

$$\tau \curvearrowright_\ell \sigma := \sum_{v \in V(\sigma)} \tau \curvearrowright_{\ell, v} \sigma,$$

where $\tau \curvearrowright_{\ell, v} \sigma$ attaches the root of $\tau$ to $v$ as the left-most child. This convention records the order of covariant differentiations. At first order,

$$F\left({}^{\bullet j}_{\bullet i}\right) = V_j \triangleright V_i = \nabla_{V_j} V_i.$$

For the manifold log-ODE construction, the essential algebraic property is the pseudo-bialgebra identity

$$\#(\alpha \star_{\mathrm{MKW}} \beta) = \#(\alpha) \circ \#(\beta),$$

where $\#$ is defined on ordered forests by total covariant derivatives. This identity holds for the MKW Hopf algebra with any smooth connection; the flat/parallel-torsion assumption is only needed when one wants to interpret the primitive structure as a post-Lie morphism.

**Example A.7** (Second-order chain rule). *For one-node trees,*

$$\bullet^j \star_{\mathrm{MKW}} \bullet^i = \bullet^j \bullet^i + \overset{\bullet}{\underset{\bullet}{\mathsf{I}}}{}^j_i.$$

*Applying # to a test function $\varphi \in C^\infty(\mathcal{M})$ gives*

$$\#(\bullet^j \star_{\mathrm{MKW}} \bullet^i)\varphi = \nabla^2\varphi(V_j, V_i) + (\nabla_{V_j} V_i)\varphi.$$

*By the covariant chain rule,*

$$\nabla^2\varphi(V_j, V_i) + (\nabla_{V_j} V_i)\varphi = V_j(V_i\varphi).$$

*Thus the grafting term $\overset{\bullet}{\underset{\bullet}{\mathsf{I}}}{}^j_i$ is exactly the correction needed for $\star_{\mathrm{MKW}}$ to represent composition of differential operators.*

### A.4.5. BRANCHED ROUGH PATHS ON $\mathcal{M}$: THE MUNTHE–KAAS–WRIGHT HOPF ALGEBRA

Let $A$ denote the alphabet of driver labels, and let $\mathrm{OF}(A)$ be the vector space spanned by $A$-decorated ordered forests. We write $\tau\sigma$ for ordered concatenation of forests and

$$[\tau_1, \ldots, \tau_k]_i$$

for the planar tree with root label $i$ and ordered children $\tau_1, \ldots, \tau_k$. Thus $\overset{\bullet^j \ \bullet^k}{\underset{\bullet_i}{\mathsf{V}}} \neq \overset{\bullet^k \ \bullet^j}{\underset{\bullet_i}{\mathsf{V}}}$.

Let $\mathcal{H}_{\mathrm{MKW}}$ denote the MKW coordinate Hopf algebra of ordered forests. Its product is the shuffle product of ordered forests, and its coproduct is the full left-admissible-cut coproduct. Branched signatures on manifolds are characters of $\mathcal{H}_{\mathrm{MKW}}$. For the log-ODE construction, we use the graded dual

$$\mathcal{H}_{\mathrm{MKW}} := \mathcal{H}^\circ_{\mathrm{MKW}} = \bigoplus_{n \geq 0} \mathcal{H}^*_{\mathrm{MKW},n}.$$

In this dual Hopf algebra, the coproduct is the cocommutative deshuffle coproduct on words of ordered forests, i.e. the sum over all order-preserving splittings into two subwords. Since $\mathcal{H}_{\mathrm{MKW}}$ is connected, graded, and cocommutative over a field of characteristic zero, Theorem A.1 gives

$$\mathcal{H}_{\mathrm{MKW}} \cong U\big(\mathrm{Prim}(\mathcal{H}_{\mathrm{MKW}})\big).$$

The primitives form a Lie algebra under the commutator of $\star_{\mathrm{MKW}}$. Equipped with the left-grafting operation, this primitive structure is the free post-Lie algebra on $A$, which is the primitive algebra underlying the manifold log-ODE construction of Kern & Lyons (2023).

The product on $\mathcal{H}_{\mathrm{MKW}}$ is the Guin–Oudom product associated with left grafting. If

$$\Delta_{\mathrm{unsh}}\alpha = \sum \alpha_{(1)} \otimes \alpha_{(2)}$$

denotes the unshuffle coproduct, then

$$\alpha \star_{\mathrm{MKW}} \beta = \sum \alpha_{(1)}\big(\alpha_{(2)} \curvearrowright_\ell \beta\big),$$

with the standard post-Lie extension of $\curvearrowright_\ell$ from trees to ordered forests. In particular, for single trees $\tau, \sigma$,

$$\tau \star_{\mathrm{MKW}} \sigma = \tau\sigma + \tau \curvearrowright_\ell \sigma.$$

For example,

$$\bullet^m \curvearrowright_\ell \overset{\bullet}{\underset{\bullet}{\mathsf{I}}}{}^j_i = \overset{\bullet^m}{\underset{\bullet_i}{\overset{\bullet^j}{\mathsf{I}}}} + \overset{\bullet^m \ \bullet^j}{\underset{\bullet_i}{\mathsf{V}}}.$$

The second term records grafting at the root of $\overset{\bullet}{\underset{\bullet}{\mathsf{I}}}{}^j_i$, where the new child is inserted as the left-most child. Hence the planar order $\overset{\bullet^m \ \bullet^j}{\underset{\bullet_i}{\mathsf{V}}}$ is distinguished from $\overset{\bullet^j \ \bullet^m}{\underset{\bullet_i}{\mathsf{V}}}$.

**Example A.8** (Smooth planarly branched signature). *Let $X^a_t = t$ and $X^b_t = t^2$ on $[0,1]$. For the smooth, ordered lift,*

$$\langle \mathbf{X}_{0,t}, \bullet^a\bullet^b \rangle = \int_{0<u<v<t} dX^a_u \, dX^b_v = \frac{2}{3}t^3,$$

*whereas*

$$\langle \mathbf{X}_{0,t}, \bullet^b \bullet^a \rangle = \int_{0<u<v<t} dX_u^b \, dX_v^a = \frac{1}{3}t^3.$$

*Consequently,*

$$\langle \mathbf{X}, \overset{a}{\underset{b}{\bullet}}\overset{b}{\bullet} \rangle = \int_0^1 \langle \mathbf{X}_{0,t}, \bullet^a \bullet^b \rangle \, dX_t^b = \frac{4}{15},$$

*while*

$$\langle \mathbf{X}, \overset{b}{\underset{b}{\bullet}}\overset{a}{\bullet} \rangle = \int_0^1 \langle \mathbf{X}_{0,t}, \bullet^b \bullet^a \rangle \, dX_t^b = \frac{2}{15}.$$

*Thus the ordered trees are distinguished by the signature. Their sum recovers the corresponding unordered coefficient $2/5$.*

**Example A.9** (Failure of ordinary planar cuts). *A naive extension of BCK to planar trees, using ordinary admissible cuts, does not represent covariant elementary differentiation. It is enough to see this at a point $y \in \mathcal{M}$. Choose local vector fields $V_i, V_j, V_m$, with $V_j(y) \neq 0$, whose first covariant derivatives at $y$ satisfy*

$$\nabla_{V_j} V_i = V_j, \qquad \nabla_{V_m} V_j = 0, \qquad \nabla_{V_m} V_i = 0, \qquad \nabla_{V_j} V_m = V_j,$$

*at $y$. All identities below are evaluated at this point. For MKW left grafting,*

$$\bullet^m \curvearrowright_\ell \overset{j}{\underset{i}{\bullet}} = \overset{m}{\underset{i}{\bullet}}\overset{j}{\bullet} + \overset{m}{\underset{i}{\bullet}}{}_j .$$

*Under the elementary differential map this gives*

$$F\left( \overset{m}{\underset{i}{\bullet}}\overset{j}{\bullet} \right) + F\left( \overset{m}{\underset{i}{\bullet}}{}_j \right) = \nabla^2 V_i(V_m, V_j) + \nabla_{\nabla_{V_m} V_j} V_i.$$

*Using*

$$\nabla^2 V_i(U, W) = \nabla_U(\nabla_W V_i) - \nabla_{\nabla_U W} V_i,$$

*we obtain*

$$\nabla^2 V_i(V_m, V_j) + \nabla_{\nabla_{V_m} V_j} V_i = \nabla_{V_m}(\nabla_{V_j} V_i) = \nabla_{V_m} V_j = 0.$$

*Thus MKW left grafting gives exactly the covariant derivative of $\nabla_{V_j} V_i$ in the $V_m$-direction.*

*By contrast, ordinary planar Connes–Kreimer cuts allow the right child of $\overset{j}{\underset{i}{\bullet}}\overset{m}{\bullet}$ to be cut independently, so the graded dual also produces the right-sibling insertion $\overset{j}{\underset{i}{\bullet}}\overset{m}{\bullet}$. Its elementary differential is*

$$F\left( \overset{j}{\underset{i}{\bullet}}\overset{m}{\bullet} \right) = \nabla^2 V_i(V_j, V_m).$$

*But*

$$\nabla^2 V_i(V_j, V_m) = \nabla_{V_j}(\nabla_{V_m} V_i) - \nabla_{\nabla_{V_j} V_m} V_i = 0 - \nabla_{V_j} V_i = -V_j.$$

*Hence the ordinary planar extension contributes a nonzero elementary differential to a composition for which the covariant chain rule gives zero.*

# B. Further Mathematical Details

## B.1. Vector Field Lift Pseudocode

In this section, we provide pseudocode for the vector field lifts of Section 3.2. For vector fields $G, U_1, \ldots, U_m$, write

$$\mathrm{TCD}_\nabla(G; U_1, \ldots, U_m) := (\nabla^m G)(U_1, \ldots, U_m),$$

where the total covariant derivative is given recursively by

$$(\nabla^0 G)(y) = G(y),$$
$$(\nabla^m G)(U_1, \ldots, U_m)(y) = \nabla_{U_1}\big((\nabla^{m-1} G)(U_2, \ldots, U_m)\big)(y)$$
$$- \sum_{r=2}^m (\nabla^{m-1} G)(U_2, \ldots, \nabla_{U_1} U_r, \ldots, U_m)(y).$$

---

**Algorithm 2** $\mathcal{H}_{\sqcup\!\sqcup}$ Vector Field Lift

**Input:** driver vector fields $\{W_i \in \Gamma(T\mathcal{M})\}_{i=0}^{d-1}$, represented in a chosen frame by coefficient functions $w_i : \mathcal{M} \to \mathbb{R}^n$, Hopf algebra $\mathcal{H} = \mathcal{H}_{\sqcup\!\sqcup}(d, N)$ with cached Lyndon basis $B$, geometry $(\mathcal{M}, \nabla)$
**Precompute (once):** Lyndon words up to depth $N$; for each length-one basis index $k$, store its letter $r_k$; for each non-letter basis index $k$, store child indices $C_k = (p, q)$ from the standard factorisation
Initialise empty cache $\{F_k\}_{k=0}^{|B|-1}$
**Memoised builder:** $\mathrm{Build}(k)$ returns cached $F_k$ if present
$C_k \leftarrow B.\mathrm{children}[k]$
**if** $C_k = ()$ **then**
   $i \leftarrow r_k$
   $F_k \leftarrow W_i$
**else**
   $(p, q) \leftarrow C_k$
   $F_p \leftarrow \mathrm{Build}(p), \quad F_q \leftarrow \mathrm{Build}(q)$
   $F_k \leftarrow [F_p, F_q]_{\mathrm{Jac}}$
     equivalently, if using torsion $T$, $F_k = \nabla_{F_p} F_q - \nabla_{F_q} F_p - T(F_p, F_q)$
**end if**
Cache and return $F_k$
**for** $k = 0$ **to** $|B| - 1$ **do**
   $F_k \leftarrow \mathrm{Build}(k)$
**end for**
**Output:** lifted fields $\{F_k\}_{k=0}^{|B|-1}$ aligned with the truncated Lyndon basis of $\mathcal{H}_{\sqcup\!\sqcup}$

---

**Algorithm 3** $\mathcal{H}_{\mathrm{GL}}, \mathcal{H}_{\mathrm{MKW}}$ Vector Field Lift

**Input:** driver vector fields $\{W_i \in \Gamma(T\mathcal{M})\}_{i=0}^{d-1}$, represented in a chosen frame by coefficient functions $w_i : \mathcal{M} \to \mathbb{R}^n$, Hopf algebra $\mathcal{H} \in \{\mathcal{H}_{\mathrm{GL}}(d, N), \mathcal{H}_{\mathrm{MKW}}(d, N)\}$ with cached rooted-tree basis $B$, geometry $(\mathcal{M}, \nabla)$
**Convention:** for $\mathcal{H}_{\mathrm{GL}}$, use the Euclidean or flat torsion-free setting so that unordered children give symmetric elementary differentials; for $\mathcal{H}_{\mathrm{MKW}}$, use ordered children and total covariant derivatives for the chosen connection $\nabla$
**Precompute (once):** enumerate canonical non-planar rooted trees for $\mathcal{H}_{\mathrm{GL}}$ or planar rooted trees for $\mathcal{H}_{\mathrm{MKW}}$ up to depth $N$; for each basis index $k$, store root colour $r_k$ and child indices $C_k$
Initialise empty cache $\{F_k\}_{k=0}^{|B|-1}$
**Memoised builder:** $\mathrm{Build}(k)$ returns cached $F_k$ if present
$C_k \leftarrow B.\mathrm{children}[k], \quad i \leftarrow r_k$
**if** $C_k = ()$ **then**
   $F_k \leftarrow W_i$
**else**
   $(p_1, \ldots, p_m) \leftarrow C_k$
   $F_{p_j} \leftarrow \mathrm{Build}(p_j)$ for $j = 1, \ldots, m$
   **if** $\mathcal{H} = \mathcal{H}_{\mathrm{GL}}$ **then**
     $F_k(y) \leftarrow D^m w_i(y)[F_{p_1}(y), \ldots, F_{p_m}(y)]$
   **else**
     $F_k(y) \leftarrow \mathrm{TCD}_\nabla(W_i; F_{p_1}, \ldots, F_{p_m})(y)$
   **end if**
**end if**
Cache and return $F_k$
**for** $k = 0$ **to** $|B| - 1$ **do**
   $F_k \leftarrow \mathrm{Build}(k)$
**end for**
**Output:** lifted fields $\{F_k\}_{k=0}^{|B|-1}$ aligned with the truncated rooted-tree basis of $\mathcal{H}_{\mathrm{GL}}$ or $\mathcal{H}_{\mathrm{MKW}}$

---

### B.2. Finite-grid role of bracket channels

This appendix formalises the finite-grid distinction between the geometric signature kernel and the proposed branched signature kernel. The claim is not that geometric signatures have a different infinite-resolution law-identifiability target. Rather, at a fixed observation grid, a geometric kernel computed from path values is a kernel on the projected path law, while our branched kernel is computed on the enhanced law containing quadratic variation and covariation coordinates.

Let $\pi_n = \{0 = t_0 < t_1 < \cdots < t_{N_n} = T\}$ be the observation grid. Define $\Omega_n := (\mathbb{R}^d)^{N_n+1}$, $\mathcal{A}_n := (\mathrm{Sym}(d))^{N_n+1}$, and $\bar{\Omega}_n := \Omega_n \times \mathcal{A}_n$. We write $\bar{x} = (x, a) \in \bar{\Omega}_n$, where $x = (x_{t_0}, \ldots, x_{t_{N_n}})$ are path values and $a = (a_{t_0}, \ldots, a_{t_{N_n}})$ are bracket values. For a semimartingale $X$, the supplied enhanced observation is $(X_{\pi_n}, A_{\pi_n})$, where $A_t := \langle X \rangle_t$. Let $\Pi : \bar{\Omega}_n \to \Omega_n$ be the projection $\Pi(x, a) = x$. Throughout, $|\cdot|$ denotes the Euclidean norm on $\mathbb{R}^d$ and the Frobenius norm on $\mathrm{Sym}(d)$.

For a kernel $k(u, v) = \langle \varphi(u), \varphi(v) \rangle_{\mathcal{K}}$, write

$$\mathrm{MMD}_k(P, Q) := \|\mathbb{E}_{U \sim P}\varphi(U) - \mathbb{E}_{V \sim Q}\varphi(V)\|_{\mathcal{K}}.$$

In the training objective, we omit the data–data term in the squared MMD since it is constant in $\theta$.

**Proposition B.1** (Projected geometric kernels and bracket visibility). *Let $k_{\mathrm{geo}}^N$ be a finite-depth geometric signature kernel computed from the piecewise-linear interpolation of $x \in \Omega_n$, with feature map $\varphi_{\mathrm{geo}}^N : \Omega_n \to \mathcal{H}_{\mathrm{br}}^N$. Extend it to enhanced observations by ignoring brackets, $\bar{k}_{\mathrm{geo}}^N((x, a), (y, b)) := k_{\mathrm{geo}}^N(x, y)$. Then, for any enhanced laws $P, Q \in \mathcal{P}(\bar{\Omega}_n)$,*

$$\mathrm{MMD}_{\bar{k}_{\mathrm{geo}}^N}(P, Q) = \mathrm{MMD}_{k_{\mathrm{geo}}^N}(\Pi_\# P, \Pi_\# Q).$$

*In particular, if $\Pi_\# P = \Pi_\# Q$, then $\mathrm{MMD}_{\bar{k}_{\mathrm{geo}}^N}(P, Q) = 0$.*

*Let $\varphi_{\mathrm{br}}^N : \bar{\Omega}_n \to \mathcal{H}_{\mathrm{br}}^N$ be the truncated branched, or bracket-augmented, feature map, and let $k_{\mathrm{br}}^N(\bar{x}, \bar{y}) = \langle \varphi_{\mathrm{br}}^N(\bar{x}), \varphi_{\mathrm{br}}^N(\bar{y}) \rangle_{\mathcal{H}_{\mathrm{br}}^N}$. Suppose that a scalar bracket statistic $b : \bar{\Omega}_n \to \mathbb{R}$ appears as a linear coordinate of this feature map: there exists $v_b \in \mathcal{H}_{\mathrm{br}}^N$ such that $b(\bar{x}) = \langle \varphi_{\mathrm{br}}^N(\bar{x}), v_b \rangle$ for all $\bar{x} \in \bar{\Omega}_n$. Then*

$$\left|\mathbb{E}_P b - \mathbb{E}_Q b\right| \leq \|v_b\|_{\mathcal{H}_{\mathrm{br}}^N} \mathrm{MMD}_{k_{\mathrm{br}}^N}(P, Q).$$

*Consequently, if $\Pi_\# P = \Pi_\# Q$ but $\mathbb{E}_P b \neq \mathbb{E}_Q b$, then the path-only geometric MMD is zero while the branched MMD is strictly positive.*

*Proof.* The lifted geometric feature map is $\bar{\varphi}_{\mathrm{geo}}^N(x, a) := \varphi_{\mathrm{geo}}^N(x) = \varphi_{\mathrm{geo}}^N(\Pi(x, a))$. Hence

$$\mathbb{E}_{\bar{X} \sim P}\bar{\varphi}_{\mathrm{geo}}^N(\bar{X}) = \mathbb{E}_{X \sim \Pi_\# P}\varphi_{\mathrm{geo}}^N(X),$$

and the same identity holds with $Q$ in place of $P$. Taking the Hilbert norm of the difference gives

$$\mathrm{MMD}_{\bar{k}_{\mathrm{geo}}^N}(P, Q) = \mathrm{MMD}_{k_{\mathrm{geo}}^N}(\Pi_\# P, \Pi_\# Q).$$

If $\Pi_\# P = \Pi_\# Q$, the right-hand side is zero.

For the branched claim, define $\mu_P := \mathbb{E}_{\bar{X} \sim P}\varphi_{\mathrm{br}}^N(\bar{X})$ and $\mu_Q := \mathbb{E}_{\bar{Y} \sim Q}\varphi_{\mathrm{br}}^N(\bar{Y})$. Since $b(\bar{x}) = \langle \varphi_{\mathrm{br}}^N(\bar{x}), v_b \rangle$,

$$\mathbb{E}_P b - \mathbb{E}_Q b = \langle \mu_P - \mu_Q, v_b \rangle.$$

Cauchy's inequality gives $\left|\mathbb{E}_P b - \mathbb{E}_Q b\right| \leq \|\mu_P - \mu_Q\| \|v_b\|$, and $\|\mu_P - \mu_Q\| = \mathrm{MMD}_{k_{\mathrm{br}}^N}(P, Q)$. If $\mathbb{E}_P b \neq \mathbb{E}_Q b$, this inequality forces $\mathrm{MMD}_{k_{\mathrm{br}}^N}(P, Q) > 0$. $\square$

This is a finite-depth statement: we only claim visibility of the bracket coordinates included in $\varphi_{\mathrm{br}}^N$, not characteristicness of $k_{\mathrm{br}}^N$ for arbitrary laws on $\bar{\Omega}_n$.

The previous proposition compares the actual geometric and branched kernels. A separate question is what happens when bracket channels are not supplied. In that case, a grid-only method can become bracket-aware only by reconstructing brackets from the observed path values.

Let $R_n : \Omega_n \to \mathcal{A}_n$ be a bracket estimator. The realised quadratic covariation estimator is the canonical example:

$$[R_n(x)]_{t_m} := \sum_{k=0}^{m-1} (x_{t_{k+1}} - x_{t_k})(x_{t_{k+1}} - x_{t_k})^\top, \qquad m = 0, \dots, N_n.$$

Define the supplied and reconstructed enhanced laws by $\bar{P}_n^{\mathrm{sup}} := \mathrm{Law}(X_{\pi_n}, A_{\pi_n})$ and $\bar{P}_n^{\mathrm{rec}} := \mathrm{Law}(X_{\pi_n}, R_n(X_{\pi_n}))$.

**Proposition B.2** (Cost of reconstructing bracket coordinates)**.** *Equip $\bar{\Omega}_n$ with $d_{\bar{\Omega}_n}((x, a), (y, b)) := \max_m |x_{t_m} - y_{t_m}| + \max_m |a_{t_m} - b_{t_m}|$. Let $\mathcal{D}_n \subseteq \bar{\Omega}_n$ contain the supports of $\bar{P}_n^{\mathrm{sup}}$ and $\bar{P}_n^{\mathrm{rec}}$. If $f : \bar{\Omega}_n \to \mathbb{R}$ is $L_f$-Lipschitz on $\mathcal{D}_n$, then*

$$|\mathbb{E}f(X_{\pi_n}, R_n(X_{\pi_n})) - \mathbb{E}f(X_{\pi_n}, A_{\pi_n})| \le L_f \,\mathbb{E} \max_m |R_n(X_{\pi_n})_{t_m} - A_{t_m}|.$$

*If $\varphi_{\mathrm{br}}^N$ is $L_N$-Lipschitz on $\mathcal{D}_n$ with respect to $d_{\bar{\Omega}_n}$, then*

$$\mathrm{MMD}_{k_{\mathrm{br}}^N}(\bar{P}_n^{\mathrm{rec}}, \bar{P}_n^{\mathrm{sup}}) \le L_N \,\mathbb{E} \max_m |R_n(X_{\pi_n})_{t_m} - A_{t_m}|.$$

*Proof.* Couple the supplied and reconstructed enhanced observations by using the same grid path $X_{\pi_n}$. Since their path coordinates are identical,

$$d_{\bar{\Omega}_n}\big((X_{\pi_n}, R_n(X_{\pi_n})), (X_{\pi_n}, A_{\pi_n})\big) = \max_m |R_n(X_{\pi_n})_{t_m} - A_{t_m}|.$$

The two coupled observations lie in $\mathcal{D}_n$ almost surely, so the Lipschitz bound for $f$ gives the first claim after taking expectations.

For the MMD bound, Jensen's inequality and Lipschitzness of $\varphi_{\mathrm{br}}^N$ on $\mathcal{D}_n$ give

$$\mathrm{MMD}_{k_{\mathrm{br}}^N}(\bar{P}_n^{\mathrm{rec}}, \bar{P}_n^{\mathrm{sup}}) \le \mathbb{E}\left\| \varphi_{\mathrm{br}}^N(X_{\pi_n}, R_n(X_{\pi_n})) - \varphi_{\mathrm{br}}^N(X_{\pi_n}, A_{\pi_n}) \right\|$$
$$\le L_N \,\mathbb{E} \max_m |R_n(X_{\pi_n})_{t_m} - A_{t_m}|.$$

$\square$

*Remark* B.3 (Reconstruction is not the geometric baseline)*.* The reconstructed law $\bar{P}_n^{\mathrm{rec}}$ is not the geometric-kernel baseline. The geometric kernel baseline simply ignores the bracket channel, as shown in Proposition B.1. The reconstructed law is a hypothetical bracket-aware baseline built from grid samples alone. Proposition B.2 shows that such a baseline pays a separate bracket-estimation error. When $R_n$ is realised quadratic covariation, this error is of $\sqrt{\Delta_n}$ scale under standard continuous-semimartingale assumptions (Jacod, 2008, Thm. 2.12(ii)). Supplying analytic bracket increments avoids this reconstruction step.

*Remark* B.4 (Scope and downstream marginal errors)*.* Geometric signatures can be law-determining in the infinite-depth limit (Chevyrev & Oberhauser, 2018; 2022). Accordingly, our claims are finite-grid claims: at a fixed observation grid, a path-only geometric kernel is a function of $\Pi_\# P$, whereas the branched kernel used in this work is a function of the enhanced law $P$ on $\bar{\Omega}_n$.

The same reconstruction term can propagate to downstream finite-dimensional statistics. If a numerical solver or observation map $F_n : \bar{\Omega}_n \to C([0, T]; \mathbb{R}^m)$ is locally Lipschitz on the enhanced observations under consideration, then for any Lipschitz statistic $\Psi : C([0, T]; \mathbb{R}^m) \to \mathbb{R}^r$ and any unit linear functional $\ell : \mathbb{R}^r \to \mathbb{R}$, the function $f = \ell \circ \Psi \circ F_n$ is Lipschitz on the same set. Proposition B.2 then gives an elementary finite-grid propagation bound for such downstream statistics.

**Proposition B.5** (Primitives lift to vector fields)**.** *Let $\mathcal{H}$ be a bialgebra and let $F : \mathcal{H} \to D$ be a pseudo-bialgebra map in the sense of Definition 2.2. If $p \in \mathcal{H}$ is primitive, i.e. $\Delta p = p \otimes 1 + 1 \otimes p$, then $F(p) \in \Gamma(TM)$.*

*Proof.* It suffices to show that $F(p)$ is a derivation of $C^\infty(M)$. Fix arbitrary $\phi, \psi \in C^\infty(M)$. By property (b) of Definition 2.2,

$$F(p)(\phi \cdot \psi) = F(\Delta p)(\psi \otimes \phi),$$

where, for simple tensors, $F(a \otimes b)(\psi \otimes \phi)$ denotes $(F(a)\psi) \cdot (F(b)\phi)$. Since $p$ is primitive,

$$F(\Delta p)(\psi \otimes \phi) = F(p \otimes 1 + 1 \otimes p)(\psi \otimes \phi)$$
$$= (F(p)\psi) \cdot F(1)\phi + F(1)\psi \cdot (F(p)\phi).$$

Property (a) gives $F(1) = \mathrm{Id}$. Hence

$$F(p)(\phi \cdot \psi) = (F(p)\psi) \cdot \phi + \psi \cdot (F(p)\phi)$$
$$= (F(p)\phi) \cdot \psi + \phi \cdot (F(p)\psi),$$

where the last equality uses commutativity of pointwise multiplication. Thus $F(p)$ satisfies the Leibniz rule, and therefore $F(p) \in \Gamma(TM)$. $\qquad\square$

## C. Further Experiments

### C.1. Recovering Log-NCDE

As stated in Section 3.4, BNRDE recovers log-NCDE when $\mathcal{H} = \mathcal{H}_{\sqcup\!\sqcup}$ and $\mathcal{M} = \mathbb{R}$. Results on PPG-DaLiA (Reiss et al., 2019) confirm this, with both models attaining an identical 0.134 mean squared error (MSE) after 10 epochs.

### C.2. Signature Computation Times

We report signature precomputation times for the relevant pySigLib log signature computations below, in Figure 7.

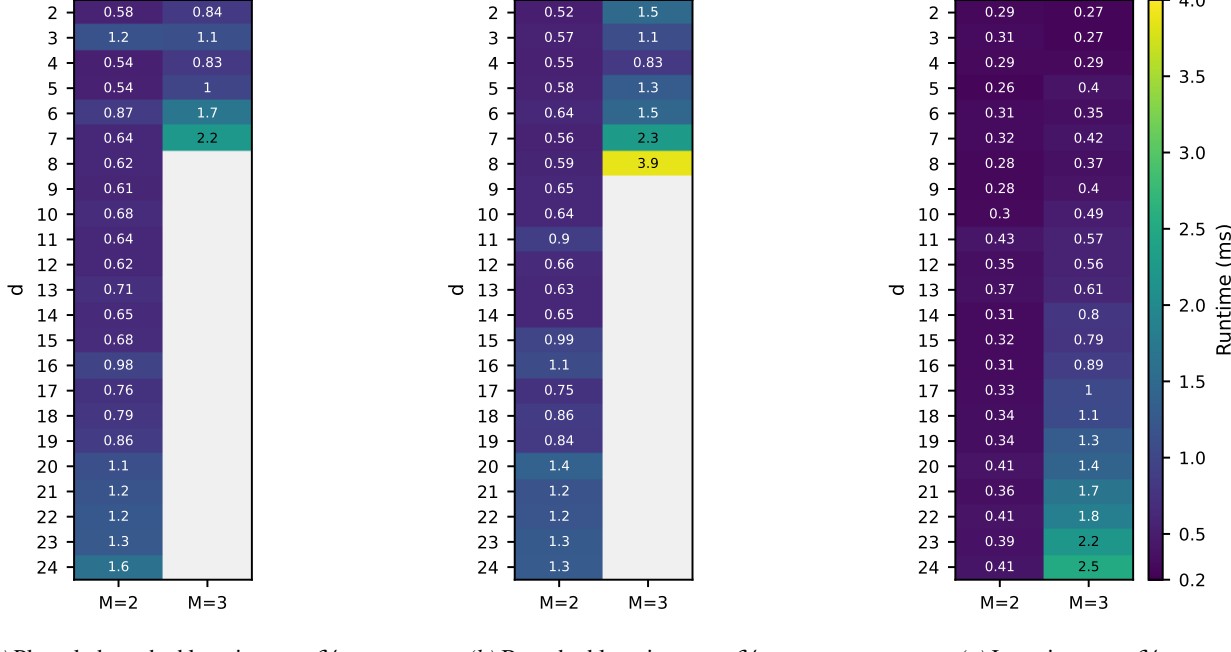

*(a)* Planarly branched log-signature $\mathcal{H}_{\mathrm{MKW}}$.     *(b)* Branched log-signature, $\mathcal{H}_{\mathrm{GL}}$.     *(c)* Log-signature, $\mathcal{H}_{\sqcup\!\sqcup}$.

*Figure 7.* Milliseconds to compute log-signatures for $\mathcal{H} \in \{\mathcal{H}_{\sqcup\!\sqcup}, \mathcal{H}_{\mathrm{GL}}, \mathcal{H}_{\mathrm{MKW}}\}$ with path dimension $d \in [2, 24]$ and signature truncation depth $N \in \{2, 3\}$. The custom CUDA kernels implemented in pySigLib do not support tree counts greater than 1024, producing the blank cells in Figure 7a and Figure 7b.

## D. Experimental Details

### D.1. Software and Hardware Details

**Software details**  All experiments were conducted on Python 3.13 using JAX 0.8.1 (Bradbury et al., 2018). We use Diffrax 0.7.1 (Kidger et al., 2021) for its Euclidean differential equation framework, Equinox 0.13.2 (Kidger & Garcia, 2021) as our neural network framework, Optax 0.2.6 (DeepMind et al., 2020) for its implementation of the Muon optimizer (Jordan et al., 2024), and Cyreal 0.1.5 (Morad, 2026) for dataloading. All $\log_{\mathcal{H}}$-signatures are formed using a custom PySigLib fork (Shmelev & Salvi, 2025), with the log-ODE method provided by Roughrax and geometric numerical integrators provided by Georax (Shmelev et al., 2026).

**Released packages**   We release two JAX packages to support the present work: one for the log-ODE method on manifolds and one containing the benchmark datasets used.

*Table 6.* New packages introduced to support the present work.

| Package | Ecosystem | Functionality | License |
|---------|-----------|---------------|---------|
| Roughrax | JAX | Euclidean and manifold log-ODE | Apache-2.0 |
| RoughBench | JAX | Rough volatility (e.g., rBergomi) and SPD diffusion benchmarks | Apache-2.0 |

**Hardware details**   All model training and evaluation was performed on an NVIDIA RTX 5080 (16GB) with CUDA 13.1, an AMD Ryzen 9 9950X3D processor running on Ubuntu 24.04 with 64GB of system memory.

### D.2. Rough Volatility

Rough Bergomi (rBergomi) is traditionally formulated as a system of coupled Volterra stochastic differential equations (SDEs) with rough kernels:

$$S_t = S_0 + \int_0^t S_s \sqrt{v_s} \, dB_s,$$

$$v_t = \xi_0(t) \exp \left( \frac{\eta}{\Gamma\left(H + \frac{1}{2}\right)} \int_0^t (t-s)^{H-\frac{1}{2}} \, dW_s - \frac{\eta^2}{2\,\Gamma\left(H + \frac{1}{2}\right)^2} \int_0^t (t-s)^{2H-1} \, ds \right).$$

Here, $S_t$ denotes the underlying asset price, $v_t$ denotes the instantaneous variance process, $\eta$ is the vol-of-vol, and $H \in (0, \frac{1}{2})$ is the Hurst exponent controlling volatility roughness. The forward variance curve $\xi_0(t)$ satisfies $\mathbb{E}[v_t] = \xi_0(t)$ and the correlation of the two drivers $W_t, B_t$ is given by $\text{Corr}(dW_t, dB_t) = \rho \in [-1, 1]$.

To simulate our rBergomi sample paths, we cast the model into the rough differential equation (RDE) framework of (Bonesini et al., 2024),

$$S_t = S_0 + \int_0^t S_u \sqrt{v_0} \exp\left\{ \nu V_u - \frac{\nu^2 u^{2H}}{2\Gamma(H + \frac{1}{2})^2} \right\} d\mathbf{W}_u - \int_0^t \frac{1}{2} S_u v_0 \exp\left\{ 2\nu V_u - \frac{\nu^2 u^{2H}}{\Gamma(H + \frac{1}{2})^2} \right\} du$$

$$V_t = \frac{1}{\Gamma(H + \frac{1}{2})} \int_0^t (t-u)^{H-\frac{1}{2}} \left( \rho dW_u + \sqrt{1 - \rho^2} dB_u \right), \tag{14}$$

where $\mathbf{W}_t$ is the signature of $W_t$ and $V_t$ is a Riemann-Liouville volatility process computed following (Bennedsen et al., 2017). We then solve Equation (14) as a Wong-Zakai ordinary differential equation (ODE) driven by a two-dimensional lead-lag path $(W_t, V_t)$, yielding convergence to the Itô solution. We use the calibration results of Callum (2023) and set $v_0 = 0.04$, $\nu = 1.991$, $H = 0.25$, $\rho = -0.848$ which shows a MSE of $3.73 \times 10^{-5}$ of model-generated implied volatilities over SPX compared to market implied volatilities on 30/05/2022. We employ the hybrid scheme of Bennedsen et al. (2017) to generate the Riemann-Liouville fractional Brownian driver.

### D.3. Sim-to-Real Dynamics Forecasting

In this section, we describe our implementation of the sim-to-real dynamics forecasting experiment developed by Bastian et al. (2025).

We begin by generating the `FREE_ROTATION` synthetic dataset available at their official repository. We then convert to `.npy` and convert the (quaternion-last) data into a timeseries of $SO(3)$ rotation matrices and form stride-one length-twenty sliding windows across the time series and flatten the $\mathbb{R}^{3\times3}$ matrices into $\mathbb{R}^9$ vectors for feeding into the models. This produces a dataset of shape $\mathbb{R}^{\text{batch}\cdot\text{damping}\times\text{windows}\cdot20\times9}$.

Each window is divided into two segments, `pred` $\in \mathbb{R}^{12\times9}$, and `recon` $\in \mathbb{R}^{8\times9}$, the ground truth `recon` is discarded, and an extrapolator fit to `pred` produces a new `recon` which is concatenated with the ground truth `pred`. We list the extrapolators here:

1. SO(3)-neural controlled differential equation (NCDE): Hermite polynomial

2. SO(3)-GRU: Hermite polynomial

3. SG-NCDE: Weighted Savitsky-Golay polynomial

4. Branched neural rough differential equation (B-NRDE): multilayer perceptron (MLP)

To save parameters in the readout MLP, all models output vectors in $\mathbb{R}^6$, which are transformed back into $\mathbb{R}^{3\times3}$ rotation matrices via a Gram-Schmidt procedure (Zhou et al., 2020).

### D.4. Mean-reverting Dynamics over the SPD Manifold under Affine-invariant Geometry

We simulate Equation (11) on $\mathbb{S}_d^{++}$ with $d = 3$ over horizon $T = 1$ on a uniform grid of $N = 1025$ time points, with initial condition $X_0 = I_3$. We set the (matrix-valued) parameter

$$\Sigma = \begin{bmatrix} 0.18 & 0.04 & -0.02 \\ 0.02 & 0.14 & 0.06 \\ 0.00 & 0.08 & 0.16 \end{bmatrix}, \qquad A = \begin{bmatrix} 0.00 & 0.20 & 0.00 \\ -0.10 & 0.00 & 0.10 \\ 0.00 & -0.15 & 0.00 \end{bmatrix}, \qquad \gamma = 0.25, \qquad \varepsilon = 0.1, \qquad \sigma = 0.4.$$

Define

$$Q := \Sigma^\top \Sigma, \qquad M := b := (d-1)Q + \varepsilon I_d, \qquad H := -\gamma I_d + A.$$

For the affine-invariant mean reversion we use the (PSD) stiffness

$$K := \Pi_{\mathbb{S}_d^+}\big(\operatorname{sym}(-H)\big), \qquad \eta := \tfrac{1}{d}\operatorname{tr}(K),$$

where $\Pi_{\mathbb{S}_d^+}$ denotes eigenvalue clipping to $\mathbb{S}_d^+$. We take the symmetric Brownian driver $B_t \in \operatorname{Sym}(d)$ with covariation

$$\mathrm{d}\langle B_{ab}, B_{cd}\rangle_t = \tfrac{1}{2}\big(\delta_{ac}\delta_{bd} + \delta_{ad}\delta_{bc}\big)\,\mathrm{d}t,$$

(corresponding to independent diagonal entries and off-diagonals scaled by $1/\sqrt{2}$), and use the identity correlation in the symmetric degrees of freedom.

**Analytic quadratic variation.**  For the Itô diffusion

$$\mathrm{d}X_t = \eta\,\log_{X_t}(M)\,\mathrm{d}t + \sigma\,X_t^{1/2}\,\mathrm{d}B_t\,X_t^{1/2},$$

the entrywise quadratic covariation of $X_t$ is

$$\mathrm{d}\langle X_{ij}, X_{mn}\rangle_t = \frac{\sigma^2}{2}\Big(X_{im}X_{jn} + X_{in}X_{jm}\Big)\,\mathrm{d}t, \qquad 1 \le i, j, m, n \le d.$$

Writing $x_t = \operatorname{vech}(X_t) \in \mathbb{R}^q$ with $q = \frac{d(d+1)}{2}$, the corresponding quadratic variation is obtained by the linear projection

$$\mathrm{d}\langle x\rangle_t = E\,\mathrm{d}\langle \operatorname{vec}(X)\rangle_t\,E^\top,$$

where $E \in \mathbb{R}^{q \times d^2}$ selects the lower-triangular coordinates consistent with our vech convention. In the generated dataset we store per-step increments $\Delta\langle x\rangle_k \approx (t_{k+1} - t_k)\big(\mathrm{d}\langle x\rangle_t/\mathrm{d}t\big)\big|_{t=t_k}$.

### D.5. Numerical Integration

For the models NCDE, log-NCDE, NRDE, and B-NRDE, which employ numerical ODE integrators, we use Tsit-5 (Tsitouras, 2011) with automatic initial stepsizing as described in Hairer et al. (2008, Sec. 2.4) and automatic iterate stepsizing using a proportional-integral-derivative controller following Söderlind (2002) and Hairer & Wanner (2002, Sec. 4.2). Backpropagation through the solver proceeds by the discretise-then-optimise method (Ma et al., 2021).

## D.6. Hyperparameters

All models are trained for 100 epochs with batch size 1024, gradient clipping at 1.0, and a 0.8/0.1/0.1 train/validation/test split. We use Muon with learning rate $5 \times 10^{-4}$, $(\beta_1, \beta_2) = (0.9, 0.999)$, weight decay $1 \times 10^{-6}$, and $\epsilon = 1 \times 10^{-8}$. NCDE-type models are solved with Tsit5 using PID tolerances $\text{rtol} = 1 \times 10^{-3}$ and $\text{atol} = 1 \times 10^{-3}$, with minimum step size $1 \times 10^{-4}$. Signature-based models use Heun with a fixed stepsize. Unless otherwise stated, models use initial-condition and vector-field MLPs of width 32, depth 2, and latent state dimension 32. Signature depth is always taken at $N = 2$. Hyperparameters differing between models are listed in Table 7.

*Table 7.* Hyperparameters that vary across experiments/models.

| Experiment | Hyperparameter | Model | | | | | | | |
|---|---|---|---|---|---|---|---|---|---|
| | | GRU | xLSTM | Stacked xLSTM | NCDE | NCDE++ | log-NCDE | NRDE | B-NRDE |
| Rough Bergomi | Activation | — | — | — | Softplus | Softplus | Softplus | Softplus | LipSwish |
| | Final activation | — | — | — | Identity | tanh | tanh | tanh | tanh |
| | Control interpolation | — | — | — | Linear | Hermite | Linear | Linear | Linear |
| | Signature window | — | — | — | — | — | 16 | 16 | 16 |
| | Hopf algebra | — | — | — | — | — | $\mathcal{H}_{\sqcup}$ | $\mathcal{H}_{\sqcup}$ | $\mathcal{H}_{\mathrm{GL}}$ |
| | Signature kernel | — | — | — | Geometric | Geometric | Geometric | Geometric | Branched |
| SO(3) Dynamics | Activation | Softplus | — | — | Softplus | Softplus | Softplus | Softplus | LipSwish |
| | Final activation | — | — | — | — | — | — | — | — |
| | Control interpolation | — | — | — | Hermite | Hermite | Hermite | Hermite | — |
| | Signature window | — | — | — | — | — | 4 | 4 | 4 |
| | Hopf algebra | — | — | — | — | — | $\mathcal{H}_{\sqcup}$ | $\mathcal{H}_{\sqcup}$ | $\mathcal{H}_{\mathrm{MKW}}$ |
| | Signature kernel | — | — | — | Geometric | Geometric | Geometric | Geometric | Branched |
| | Extrapolator | — | — | — | Hermite | Hermite, SG | Hermite | Hermite | MLP |
| SPD Dynamics | Activation | — | — | — | — | — | — | — | — |
| | Final activation | — | — | — | — | — | — | — | — |
| | Control interpolation | — | — | — | — | — | — | — | — |
| | Signature window | — | — | — | — | — | 64 | 64 | 64 |
| | Hopf algebra | — | — | — | — | — | $\mathcal{H}_{\sqcup}$ | $\mathcal{H}_{\sqcup}$ | $\mathcal{H}_{\mathrm{MKW}}$ |
| | Signature kernel | — | — | — | — | — | Geometric | Geometric | Branched |

