# OpenReview forum: "Learning Manifold and Itô Dynamics with Branched Neural Rough Differential Equations"
_ICML.cc/2026/Conference — ICML 2026 regular_

### Official Review · Reviewer_RQRn · 2026-03-08

**Soundness:** 3
**Presentation:** 2
**Significance:** 3
**Originality:** 3
**Overall Recommendation:** 5
**Confidence:** 2

**Summary:**

This paper proposes a novel Branched Neural Rough Differential Equation network to introduce stochasticity and manifold constraints to the NRDE. The paper is well-written but some discussions and experiments are insufficient.

**Compliance With Llm Reviewing Policy:**

Affirmed.

**Final Justification:**

The paper is well-written and my concerns are addressed. To summarize, the soundness, originality, and clarity are high and the significance is moderate as the experiments mostly focus on manifold constrained cases. However, in general, the paper is novel. The authors have addressed my concerns. In general, I suggest "5: Accept: Technically solid paper".

**Key Questions For Authors:**

1.	Some concepts, like Ito, MKW, Stratonovich, etc., should be roughly explained when first used.
2.	It’s hard to connect preliminary 2.1 and 2.2/2.3. In particular, what’s the disadvantage of preliminary 2.1, and how fundamentals in 2.2/2.3 have the potential to address these concerns?
3.	The author is suggested to test some more common tasks, e.g., irregularly sampled time series.
4.	Some baselines that consider computing neural ODEs on manifolds should also be stated and tested.
5.	Some ablation studies, sensitivity analyses, and computational time should be investigated.

**Limitations:**

yes

**Strengths And Weaknesses:**

The paper is mathematically sound and well-written. However, more analyses and experiments are needed.

---

> ### Author Rebuttal · Authors · 2026-03-30
>
> [1] **Some concepts, like Itô, MKW, Stratonovich, etc., should be roughly explained when first used**
>
> We agree that these concepts should be introduced more clearly, and will revise the manuscript accordingly.
> - **Stratonovich.** Now introduced as the stochastic midpoint integral.
> - **Itô.** Now introduced as the non-anticipative stochastic integral for semimartingales.
> - **GL.** Now described as the tree algebra, whose extra coordinates encode quadratic variation terms.
> - **MKW.** Now described as the ordered-tree analogue of GL, needed on manifolds where successive motions do not commute.
>
> [2] **It’s hard to connect preliminary 2.1 and 2.2/2.3. In particular, what’s the disadvantage of preliminary 2.1, and how fundamentals in 2.2/2.3 have the potential to address these concerns?**
>
> We agree this transition was not sufficiently explicit.
> - Section 2.1 is the familiar geometric/Euclidean starting point, not a flawed formulation. Its limitation is that it covers the classical log-signature viewpoint, but not the algebraic structure later needed for branched Itô terms and order-sensitive manifold terms.
> - Section 2.2 introduces Hopf algebras as the combinatorial language for these richer signatures.
> - Section 2.3 then shows how these algebraic objects act concretely through vector fields via pseudo-bialgebra maps.
>
> Our revised manuscript now makes this progression more explicit. The start of Section 2.3 states: "Pseudo-bialgebra maps make the action of signatures through vector fields explicit by representing both the signature and the induced differential operators in a common word- or tree-based basis."
>
> [3] **The author is suggested to test some more common tasks, e.g., irregularly sampled time series**
> - We agree that more common time series tasks would strengthen the paper. While we were not able to add an irregularly sampled benchmark within the rebuttal window, we added a Euclidean long-sequence benchmark (PPG-DaLiA, length 49920) in the appendix.
> - This does not fully substitute for the irregular-sampling setting requested by the reviewer, but it does test whether the broader BNRDE framework loses performance in a standard Euclidean regime. In this setting, BNRDE recovers log-NCDE as a special case and matches its performance exactly.
>
> | Model | MSE |
> | - | - |
> | NCDE         | 0.160 |
> | log-NCDE     | 0.134 |
> | BNRDE        | 0.134 |
>
> We will add irregularly sampled time series benchmarks in the revised manuscript.
>
> [4] **Some baselines that consider computing neural ODEs on manifolds should also be stated and tested**
>
> We have added manifold neural ODE (MNODE) ([Lou et al. (2020)](https://arxiv.org/abs/2006.10254)) to Section 4.3.
>
> | Method | Pretraining Frobenius | Static Motion | Translation Motion | Unconstrained Motion |
> |-|-:|-:|-:|-:|
> | MNODE | 2.461 | 132.40 | 118.25 | 121.08 |
> | BNRDE | **0.049** | _3.23_ | _3.70_ | _3.33_ |
>
> - MNODE performs substantially worse, which is consistent with the task: in this chaotic forecasting problem, conditioning only on the initial state is much less informative than conditioning on a window of observed trajectory.
> - We do not believe neural ODE is an appropriate baseline for Sections 4.2 and 4.4. In those experiments, the initial condition is fixed while randomness enters through the driving noise, so an ODE baseline would produce the same trajectory each time.
> - We also enriched Section 4.4 with NRDE, NCDE, and NCDE++ baselines. BNRDE retains leading performance.
>
> [5] **Some ablation studies, sensitivity analyses, and computational time should be investigated.**
> - We added runtime and memory plots across depths $N = 1,2,3,4$ and path dimensions $d = 1,2,4,8,12,16,20,24$. They are available here: https://imglink.cc/cdn/6kggrbSU-o.png, https://imglink.cc/cdn/EYWcrDB-sV.png, https://imglink.cc/cdn/vi0rdf4PKl.png
> - We added ablations comparing the standard (GK) and branched (BK) signature kernels on experiments 4.2 and 4.4. The branched signature kernel improves the KS score over the standard signature kernel, in line with Remark 4.1 and Theorem B.2. The Section 4.2 results are:
>
> | Model | Training time | 128 | 256 | 384 | 512 |
> | - | - | - | - | - | - |
> | BNRDE (GK)     | 926 | 10.36 ± 1.35 | 8.53 ± 1.25 | 7.94 ± 1.06 | **5.84 ± 1.62** |
> | **BNRDE (BK)** | 927 | **6.50 ± 1.97** | **5.56 ± 1.35** | **7.14 ± 0.73** | 7.91 ± 1.63 |
>
> - We will also analyse sensitivity to the ODE solver order in the revision, e.g., Heun versus Tsit5.
> - We also conducted depth sensitivity analysis; we show only rBergomi due to word limits. We do not observe a consistent improvement from depth >2.
>
> | Depth | 128 | 256 | 384 | 512 |
> |-|-|-|-|-|
> | 2 | 0.06 ± 0.02 | 0.06 ± 0.01 | 0.07 ± 0.01 | 0.08 ± 0.02 |
> | 3 | 0.07 ± 0.01 | 0.06 ± 0.01 | 0.09 ± 0.01 | 0.10 \pm 0.00 |
>
> In the revised manuscript, we will also consider signature length (for 4.2 and 4.4; it is necessarily fixed for 4.3) and roughness sensitivity (varying the Hurst parameter in 4.2).

---

> > ### Author Rebuttal · Reviewer_RQRn · 2026-04-01
> >
> > Thanks for the replies. I will consider rasing the score.

---

> > > ### Author Response · Authors · 2026-04-03
> > >
> > > Thank you for considering our response. We are pleased that our clarifications have helped address your concerns. We would be grateful if you would consider reflecting this in your score.

---

### Official Review · Reviewer_tMgp · 2026-03-13

**Soundness:** 3
**Presentation:** 4
**Significance:** 3
**Originality:** 3
**Overall Recommendation:** 5
**Confidence:** 3

**Summary:**

This paper modifies neural rough differential equations (NRDEs) to address two shortcomings: (i) stochastic dynamics in the Ito sense, and (ii) dynamics evolving on manifolds where the curvature renders the effect of repeated derivatives _order-dependent_. The authors propose branched NRDEs to create models that are well-defined under Ito noise and on manifolds, by using tree-based rough path lifts (over rough path Hopf algebras) replacing geometric signatures which is a standard tool in rough path that summarizes the statistics/features of the rough path. The authors also introduce an Ito-consistent training objective using these branched signatures. An autodiff Jax package Stochastax is provided, along with experiments across forecasting and volatility modeling.
The authors also unify the algebraic approaches and convert the structures into a palatable framework of learned vector fields and differential operators on a target manifold.

**Compliance With Llm Reviewing Policy:**

Affirmed.

**Key Questions For Authors:**

Please see the Strengths and Weaknesses section above. I am willing to raise my score if things are clarified to improve my understanding of the problem. Thank you.

**Limitations:**

The authors acknowledge the $ d^2$ limitation coming from the truncated iterated-integrals;
And I just want to make sure, as previously stated, that the end-to-end claims put forth by the authors don't have gaps (in **Weaknesses**).

**Strengths And Weaknesses:**

** Strengths**
* There is an interesting extension of NRDEs to the case where the shuffle algebra breaks down. The problem formulation is clear, and well-motivated. The use of branching from earlier rough paths theory is a novel thing to incorporate into this NRDE setting.
* The unifying algebraic viewpoint is valuable, especially the introduction of the log-ODE and bialgebra framework to translate.
* Experiments are not favorable to Euclidean NRDEs, which shows the strength of the authors' methodology and training objective. Generally, the experiments seem nontrivial and aligned with the theory put forth.

**Weaknesses**
* It isn't clear to me how the signature kernel objective consistently learned the right Ito law.
* I'm not sure if the truncated model directly inherits the other desiderata.
* What is the error based on the tangent space projection - retraction method in Section 3.5? This is a pretty big modeling approximation, and I'm not sure how it works over highly curved regions or long horizons.
* The experiments, while nontrivial, don't really show the validation of theoretical claims (the SO(3) task uses $H_{shuffle}$ as the geometry, not the branched manifold case - would that apply?).

---

> ### Author Rebuttal · Authors · 2026-03-29
>
> We thank the reviewer for the positive comment and we address the concerns as follows.
>
> [1] **It isn't clear to me how the (branched) signature kernel objective consistently learned the right Ito law.**
> * We did not intend to claim convergence to a different underlying law of the observed process. Rather, the branched signature kernel works on the Itô/branched enhancement of the path, which is the representation in which quadratic-variation effects are explicit. Our theorem shows that, for geometric (shuffle) signature kernels, the error decomposes into the usual rough discretisation term plus an additional quadratic variation estimation error. The advantage of the branched formulation is that it targets the correct Itô-enhanced object directly, thereby avoiding the need to recover this quadratic variation contribution indirectly (like by squared increments).
> * We acknowledge this was not made clear in the manuscript, and have updated the last sentences of the branched kernel section to more clearly connect it to our theorem.
>
> [2] **I'm not sure if the truncated model directly inherits the other desiderata**
> * To demonstrate this empirically, we have expanded our rBergomi (Section 4.2) experiments by adding a B-NRDE trained with the standard (shuffle) signature kernel (GK) and our branched signature kernel (BK). The BK model outperforms the GK model, evidencing our our claim in remark 4.1 and Theorem B.2 that the branched kernel improves convergence to the true time-marginal law (resulting in lower KS discrepancies).
> | Model | Training time | 128 | 256 | 384 | 512 |
> | --- | --- | --- | --- | --- | --- |
> | BNRDE (GK) | 926 | 10.36 ± 1.35 | 8.53 ± 1.25 | 7.94 ± 1.06 | **5.84 ± 1.62** |
> | **BNRDE (BK)** | 927 | **6.50 ± 1.97** | **5.56 ± 1.35** | **7.14 ± 0.73** | 7.91 ± 1.63 |
> * We will also add GK experiments to 4.4 in case of acceptance
>
> [3] **What is the error based on the tangent space projection - retraction method in Section 3.5? ...**
> * We thank the reviewer for raising this point. Section 3.5 uses a single fixed tangent space $T_{Z_0}M$ for the entire solve, so it is certainly a computational approximation. Accordingly, we do not claim uniform accuracy over highly curved regions or very long horizons. While the trajectory remains in one retraction neighbourhood, subtracting the intrinsic pulled-back ODE and the fixed-chart ODE yields an inequality of the form
>  $$\frac{d}{dt} \mathrm{error}(t) \le L \mathrm{error}(t) + \delta(t),$$
>  where $L$ is a local Lipschitz constant and $\delta(t)$ is the chart/retraction defect. Applying Gronwall then gives
>  $$\mathrm{error}(t) \le e^{Lt}\int_0^t \delta(s)  ds.$$
>  Thus the approximation is local and problem-dependent, and can deteriorate when the trajectory moves far from Z_0, when curvature is large, or when the horizon is long. This problem dependence means we cannot give a clean theorem which describes the error for all problems applicable to BNRDE.
> * To limit this error, one may work in more than one tangent space by restarting the log-ODE step with the previous endpoint as the new initial condition. We will describe this idea in the manuscript, and how it can reduce error accumulation.
>
> [4] **The experiments, while nontrivial, don't really show the validation of theoretical claims (the SO(3) task uses the shuffle Hopf algebra as the geometry, not the branched manifold case - would that apply?).**
> * We do not believe the deterministic SO(3) task is the natural setting to isolate the benefits of the branched construction: for deterministic controls, quadratic covariation vanishes, so the additional branched coordinates are trivial. Instead, the Stratonovich iterated integrals of the shuffle algebra, which reduce to standard Riemann integrals in this deterministic case, is the canonical choice.
> * Accordingly, we do not present this experiment as validation of the specifically branched-manifold theory. Its purpose is instead to validate a separate theoretical claim of the paper, namely the extension of shuffle-Hopf neural RDEs to manifold-valued dynamics via the unifying idea of vector field lifts. The subsequent SPD manifold SDE experiment is the one intended to validate the genuinely branched rough-path regime on manifolds.
>
> [5] **$d^2$ Limitations**
>
> We will extend discussion of the $d^2$ scaling limitation with these additions:
> * We have added a computational complexity section which derives FLOPS cost for each Hopf algebra at a given depth N. This connects how the $d^2$ scaling relates to the number of basis elements of the GL and MKW tree algebras.
> We have added training and inference time results for each experiment to the appendix. In addition, we now provide the time and memory cost of computing the signatures and vector field lifts under each Hopf algebra across depths 1, 2, 3, 4 for path dims 1, 2, 4, 8, 12, 16, 20, 24.
> * We believe these additions clarify this limitation, and provide strong motivation for future work in neural RDE.

---

> > ### Author Rebuttal · Reviewer_tMgp · 2026-04-04
> >
> > Thanks for the clarifications, I have a better understanding of the work and its motivation now.

---

> > > ### Author Response · Authors · 2026-04-04
> > >
> > > We thank the reviewer for their constructive remarks, and hope our clarifications and additions have improved their confidence in our work.

---

### Official Review · Reviewer_3qeW · 2026-03-13

**Soundness:** 4
**Presentation:** 3
**Significance:** 3
**Originality:** 3
**Overall Recommendation:** 4
**Confidence:** 2

**Summary:**

This paper introduces Branching Neural Rough Differential Equations (B-NRDEs). To overcome the limitations of the standard NRDE, namely its inability to handle the quadratic variation of Itoh integrals and the noncommutativity problem caused by manifold curvature, the authors replace the traditional shuffle algebra with tree algebras. Through pseudo-bialgebra maps, these abstract tree structures are unified and transformed into vector fields computable by neural networks; furthermore, the authors have open-sourced Stochastax, a JAX library that supports automatic differentiation.

**Compliance With Llm Reviewing Policy:**

Affirmed.

**Final Justification:**

Thanks the reviewer for the second round discussion, and I will maintain my positive score to support this work as the rebuttal phase address my concerns.

**Key Questions For Authors:**

**Questions:**
1. Could the authors provide time and memory benchmarks across varying dimensions $d$ and depths $N$ to quantify the $O(d^2)$ bottleneck impact?
2. Can initial condition enhancements mitigate the severe starting point deviations in SPD experiments?

**Limitations:**

yes

**Strengths And Weaknesses:**

**Strengths:**
1. Unifies Itô noise and manifold curvature through non-commutativity, using tree structures and pseudo-bialgebraic mappings.
2. Natively supports quadratic variation in financial Itô modeling, eliminating the feature dimension doubling from lead-lag lifting.
3. The theoretical connection to Neural SPDEs outlines a highly promising and feasible direction for future continuous-time dynamics research.

**Weaknesses:**
1. Quadratic variation computation scales as $O(d^2)$, limiting models to low-dimensional states and preventing high-dimensional applications.
2. SO(3) task performance is inferior to simpler SG-NCDE baseline. SPD manifold experiments show poor trajectory fidelity near starting points due to signature translation invariance.

---

> ### Author Rebuttal · Authors · 2026-03-31
>
> We thank the reviewer for their feedback on our work, particularly the idea to examine initial condition modifications, which resulted in an uplift in model performance.
>
> [1] **Could the authors provide time and memory benchmarks across varying dimensions and depths to quantify the $d^2$ bottleneck impact?**
> - We agree that benchmarking the combinatorial growth of tree-based representations and the associated $d^2$ scaling is an important metric in assessing the practical feasibility of our method. To address this, we have added runtime and memory plots for signature generation and vector-field lifts across depths $N=1,2,3,4$ and path dimensions $d=1,2,4,8,12,16,20,24$. They are available here: https://imglink.cc/cdn/6kggrbSU-o.png, https://imglink.cc/cdn/EYWcrDB-sV.png, https://imglink.cc/cdn/vi0rdf4PKl.png
> - In terms of full model training, we benchmark up to $d=16$, finding:
>
> | algebra | dim | train (s) | inference (s) |
> |:-|-:|-:|-:|
> | shuffle |   2 |  23.7 | 0.14 |
> | shuffle |   4 |  24.9 | 0.13 |
> | shuffle |   8 |  57 | 0.15 |
> | shuffle |  16 |  168 | 0.16 |
> | GL      |   2 |   23.7 | 0.14 |
> | GL |   4 |  25.3 | 0.14 |
> | GL      |  8|  41.5 | 0.15 |
> | GL      |  16 |  252 | 0.16 |
> | MKW     |   2 |  23.7 | 0.14 |
> | MKW |   4 |  25.5 | 0.14 |
> | MKW |   8 |  40.6 | 0.15 |
> | MKW     |  16 |  251 | 0.16 |
> * We are also running higher-depth benchmarks and will report them in the revision. Our current runtime/memory curves already make clear that higher depths are substantially more challenging, especially for MKW. However, we do not consider a serious blocker as we find depth 2 sufficient for the experiments consdiered (see [5])
> - We will therefore revise the manuscript to make the scaling claim more precise: BNRDE is practical for moderate-dimensional real-world systems in the regime used in the paper, but we do not claim arbitrary scalability to large $d$ and large $N$.
> - We have also added new theoretical analyses connecting the algebra structure directly to this scaling, and showing that in the shuffle case our complexity matches established log-NCDE complexity ([Walker, et al. (2024)](https://arxiv.org/abs/2402.18512)).
>
> [2] **SO(3) task performance is inferior to simpler SG-NCDE baseline.**
> - We agree that SG-NCDE is the strongest baseline on this task. This SO(3) benchmark is a smooth, deterministic forecasting problem and therefore closely matches the setting SG-NCDE was designed for, with architecture choices tailored specifically to SO(3)
> - By contrast, B-NRDE is a general-purpose manifold model intended to handle arbitrary manifolds and rough dynamics, not only this specialised smooth setting.
> - We will clarify that the purpose of this experiment is therefore not to outperform a task-specific method on its home benchmark, but to show that B-NRDE remains competitive even in this favourable setting for SG-NCDE.
>
> [3] **Can initial condition enhancements mitigate the severe starting point deviations in SPD experiments?**
> - We tried the method introduced by [Morill, et al. (2020)](https://arxiv.org/abs/2006.00873) wherein 0 is appended to the start of the time series resulting in the first increment being the value of the initial condition (IC). This reduced IC error and loss at $t=128$ ($7.57 \to 7.11$). The plot is visible here https://imglink.cc/cdn/zw_XS4Dr-B.png.
> - Some IC deviations remain, we believe, due to the kernel loss (over all timesteps) dominating timestep-one (IC) loss, making it hard to identify
> - We also tried auxilliary IC MSE and Frobenius losses on the first timestep, however, neither were effective, worsening t=128 performance by more than 0.5 KS score, making it hard to justify their inclusion.
> - In the updated manuscript, we will add an initial-condition technique ablation using an initial condition loss (eigenvalue MSE at time 0).

---

> > ### Author Rebuttal · Reviewer_3qeW · 2026-04-04
> >
> > I appreciate the authors' detailed rebuttal, including their transparent acknowledgment of the $\mathcal{O}(d^2)$ scalability bottleneck and the addition of further baselines. However, while the proposed method theoretically achieves an elegant unification of Itô dynamics and manifold constraints, the rebuttal simultaneously confirms that the approach is subject to severe practical limitations, most notably initial condition fidelity and long-term trajectory drift. Given these inherent constraints, I will maintain my original rating.

---

> > > ### Author Response · Authors · 2026-04-04
> > >
> > > We thank the reviewer for their follow-up. We agree that initial-condition fidelity and long-horizon drift are relevant practical considerations. We would like to take this opportunity to contextualize these weaknesses in the literature and with respect to other baselines.
> > >
> > > **Initial condition mismatch**
> > > - This issue is not unique to BNRDE, occurring also in NRDE and log-NCDE, along with any other models which take the path signature as input.
> > > - We have updated our manuscript to discuss this limitation more openly.
> > > - We do not believe this to be a major practical blocker, as BNRDE remains SOTA with respect to path fidelity and time-marginal laws, which are traditionally the quantities of interest in downstream tasks (e.g., options pricing; S4.2, trajectory forecasting; S4.3).
> > >
> > > **Tangent space approximation**
> > > - We believe the fair comparison is against stochastic baselines that do not enforce the manifold geometry at all. Relative to these models, the limitation of B-NRDE is narrower: it may incur some bounded drift, while still respecting the geometry, versus alternatives that make few geometric considerations.
> > > - We also remark that if the hidden state is constrained to the same space as the input, there is no drift as a retraction may be taken after each step, rather than at the end.
> > > - In our updated manuscript, we will explicitly discuss this issue and the relevant bound described in our response to Reviewer tMgp and derive a related bound for the case in which the hidden state matches the problem geometry.
> > >
> > > We thank the reviewer for their consideration of our work, and believe these clarifications and discussions have enhanced our manuscript.

---

### Official Review · Reviewer_LUZW · 2026-03-23

**Soundness:** 3
**Presentation:** 2
**Significance:** 1
**Originality:** 3
**Overall Recommendation:** 4
**Confidence:** 3

**Summary:**

The paper introduces Branched Neural Rough Differential Equations (B-NRDE), a framework that extends neural rough differential equations to handle stochastic Itô dynamics and manifold-valued systems where classical signature methods break down. It replaces standard geometric signatures with tree-based rough path lifts derived from Hopf algebras, enabling consistent modeling of curvature and stochastic effects. Experiments across financial modeling and geometric dynamics tasks demonstrate strong performance, supported by a new auto differentiable library for efficient computation of branched signatures and RDEs.

**Compliance With Llm Reviewing Policy:**

Affirmed.

**Final Justification:**

The authors have provided a strong rebuttal with substantially improved empirical evidence.

**Key Questions For Authors:**

* How sensitive is the model performance with respect to signature truncation depth? Any way to adaptively select the depth?

**Limitations:**

Yes

**Strengths And Weaknesses:**

-Soundness-
* The paper has a strong motivation and clearly stated limitations of previous standard NRDEs.
* The proposed method well aligns with rough path theory and Hopf algebra structures. Provided claims look mathematically sound.
* The complexity of tree based representations grow combinatorially with increased depth and dimension. How much is the proposed method scalable with respect to state dimension d and truncation order N? Can proposed method handle high-dimensional real world systems beyond OMD dataset?

-Presentation-
* The paper is well organized and easy to follow.
* Impact statement is missing.

-Significance-
* The proposed method enables Ito-consistent neural dynamics to be trained beyond Euclidean space.
* The experiment section is focused on regimes where standard Euclidean NRDEs struggle. More empirical results in general long sequence prediction tasks are needed, e.g., UEA multivariate time series classification archive (UEA-MTSCA), and PPG-DaLiA, multivariate time series regression dataset. Also, more comparison models can be added such as SOTA stacked recurrent models.

-Originality-
* The authors first integrated branched rough paths into neural differential equations.
* The Jax package for auto differentiable solution for RDEs is implemented and shared.

---

> ### Author Rebuttal · Authors · 2026-03-30
>
> We thank the reviewer for their constructive comments and suggestions.
>
> [1] **The complexity of tree based representations grow combinatorially with increased depth and dimension. How scalable is the proposed method scalable w.r.t state dimension d and truncation order N? Can proposed method handle high-dimensional real world systems beyond OMD dataset?**
> - We agree that the combinatorial growth in tree-based representations should be quantified. To address this, we have added runtime and memory plots for signature generation and vector-field lifts across depths $N=1,2,3,4$ and path dimensions $d=1,2,4,8,12,16,20,24$. They are available here: https://imglink.cc/cdn/6kggrbSU-o.png, https://imglink.cc/cdn/EYWcrDB-sV.png, https://imglink.cc/cdn/vi0rdf4PKl.png
> - In terms of full model training, we benchmark up to $d=16$, finding:
>
> | algebra | dim | train (s) | inference (s) |
> |:-|-:|-:|-:|
> | shuffle |   2 |  23.7 | 0.14 |
> | shuffle |  16 |  168 | 0.16 |
> | GL      |   2 |   23.7 | 0.14 |
> | GL      |  16 |  252 | 0.16 |
> | MKW     |   2 |  23.7 | 0.14 |
> | MKW     |  16 |  251 | 0.16 |
> * Please see expanded table in our response to reviewer 2
> * We are also running higher-depth benchmarks and will report them in the revision. Our current runtime/memory curves already make clear that higher depths are substantially more challenging, especially for MKW. However, we do not consider a serious blocker as we find depth 2 sufficient for the experiments consdiered (see [5])
> - We will therefore revise the manuscript to make the scaling claim more precise: BNRDE is practical for moderate-dimensional real-world systems in the regime used in the paper, but we do not claim arbitrary scalability to large $d$ and large $N$.
> - We have also added new theoretical analyses connecting the algebra structure directly to this scaling, and showing that in the shuffle case our complexity matches established log-NCDE complexity ([Walker, et al. (2024)](https://arxiv.org/abs/2402.18512)).
>
> [2] **The experiment section is focused on regimes where standard Euclidean NRDEs struggle. More empirical results in general long sequence prediction tasks are needed, e.g., UEA multivariate time series classification archive (UEA-MTSCA), and PPG-DaLiA, multivariate time series regression dataset.**
> * We thank the reviewer for this suggestion. In the Euclidean deterministic setting of PPG-DaLiA and UEA, our method recovers log-NCDE as a special case (with $\mathcal{H}=$ shuffle and manifold $=\mathbb{R}$). The purpose of these additional experiments is therefore not to claim a new Euclidean SOTA, but to verify that the broader BNRDE framework does not lose performance in the classical setting while covering substantially more general regimes. On PPG-DaLiA, BNRDE and log-NCDE attain identical performance, and we will add UEA classification results to the revised manuscript as well.
>
> | Model | MSE |
> | - | - |
> | log-NCDE     | 0.134     |
> | BNRDE        | 0.134     |
> * We believe this directly supports the idea that BNRDE generalizes existing Euclidean methods without sacrificing performance on standard Euclidean benchmarks.
>
> [3] **More comparison models can be added such as SOTA stacked recurrent models.**
> - We agree this comparison would strengthen the empirical section. During rebuttal we were able to add NCDE/NCDE++/NRDE and a manifold neural ODE baseline, which already broadens the comparison substantially, but given the time constraint, we were not able to complete additional stacked recurrent baselines within the rebuttal window. We will add these in the revision.
>
> [4] **Impact statement is missing**
> - We apologize for this omission and will add the impact statement in the updated manuscript.
>
> ## Questions
> [5] **How sensitive is the model performance with respect to signature truncation depth?**
> - Performance is not highly sensitive to truncation depth in the tested regime: on rBergomi, $N=2,3$ are within 2SD across all reported widths, so we do not observe a significant benefit from going beyond depth 2.
>
> | Depth | 128 | 256 | 384 | 512 |
> |-|-|-|-|-|
> | Depth 2 | $0.06 \pm 0.02$ | $0.06 \pm 0.01$ | $0.07 \pm 0.01$ | $0.08 \pm 0.02$ |
> | Depth 3 | $0.07 \pm 0.01$ | $0.06 \pm 0.01$ | $0.09 \pm 0.01$ | $0.10 \pm 0.00$ |
> - We will further enrich this sensitivity analysis in the revision by also considering signature length (for Sections 4.2 and 4.4; it is necessarily fixed in 4.3) and roughness sensitivity (varying the Hurst parameter in 4.2).
>
> [6] **Any way to adaptively select the depth?**
> - Yes in principle: adaptive Log-ODE methods (e.g., [Bayer, et al. (2023)](https://arxiv.org/abs/2307.12590)) select either a smaller step or a higher truncation depth from local error estimates. Extending this to our setting requires a manifold version of these error estimates together with control of the additional geometric error introduced by tangent-space propagation/retraction, which we leave to future work.

---

> > ### Author Rebuttal · Reviewer_LUZW · 2026-04-04
> >
> > I appreciate the authors for their response. Additional empirical results truly strengthen the paper much better. I will raise the score.

---

> > > ### Author Response · Authors · 2026-04-04
> > >
> > > We thank the reviewer for acknowledging our rebuttal. During the rebuttal period, we added the suggested **stacked RNN baselines** [stacked-XLSTM, XLSTM](https://arxiv.org/abs/2405.04517) (additionally, this is highly similar to the backbone of [XLSTM-Time](https://arxiv.org/abs/2407.10240)) and LSTM, and found that **BNRDE still performs best**. We also thank the reviewer for this suggestion, as the strong stacked-RNN results further clarify the strength of BNRDE's manifold modelling.
> > >
> > > ## New baselines
> > > **rBergomi:**
> > > xLSTM, LSTM done, but excluded due to length limits
> > > | Depth | 128 | 256 | 384 | 512 |
> > > |---|---:|---:|---:|---:|
> > > | **Stacked xLSTM** | 22.31 ± 1.50 | 16.16 ± 0.58 | 14.02 ± 0.78 | 13.28 ± 1.62 |
> > > | **BNRDE** | **6.50 ± 1.97** | **5.56 ± 1.35** | **7.14 ± 0.73** | *7.91 ± 1.63* |
> > >
> > > **SO(3) experiment:**
> > > xLSTM, LSTM also done, but excluded due to length limits
> > > | Method | Pretraining Frobenius | Static Motion | Translation Motion | Unconstrained Motion |
> > > |-|-:|-:|-:|-:|
> > > | Manifold neural ODE | 2.461 | 132.40 | 118.25 | 121.08 |
> > > | Stacked xLSTM | 0.110 | 7.11 | 7.50 | 7.03 |
> > > | **BNRDE** | **0.049** | _3.23_ | _3.70_ | _3.33_ |
> > >
> > > **SPD experiment:**
> > > Confirms SOTA over other CDE/RDE-based models and stacked-RNN/XLSTM baselines
> > > | Method | 128 | 256 | 384 | 512 |
> > > |-|-:|-:|-:|-:|
> > > | NCDE++ | 28.65 ± 1.99 | 25.13 ± 3.13 | 22.88 ± 1.64 | 22.53 ± 0.94 |
> > > | NCDE | 28.85 ± 1.99 | 25.30 ± 3.90 | 22.33 ± 1.84 | 21.73 ± 1.52 |
> > > | **LSTM** | 14.32 ± 0.18 | 16.73 ± 0.24 | 18.96 ± 0.12 | 20.63 ± 0.17 |
> > > | NRDE | 10.98 ± 0.11 | 14.73 ± 0.21 | 16.49 ± 0.23 | 17.28 ± 0.33 |
> > > | **xLSTM** | 10.20 ± 0.45 | 12.28 ± 0.73 | 14.21 ± 0.89 | 14.94 ± 1.11 |
> > > | log-NCDE | 9.24 ± 0.39 | 12.14 ± 0.61 | 13.81 ± 0.40 | 14.08 ± 0.73 |
> > > | **Stacked xLSTM** | **6.97 ± 0.38** | *10.44 ± 0.90* | *12.46 ± 0.82* | *13.24 ± 0.96* |
> > > | **BNRDE (BK)** | *7.57 ± 0.24* | **9.71 ± 0.67** | **10.73 ± 0.45** | **10.78 ± 0.81** |
> > >
> > > ## Benchmarks
> > > **Signature/VF computation benchmarks:**
> > > 28 additional benchmarks clarify the cost of vector field and signature evaluations, showing they are not a bottleneck. We additionally report confidence intervals for each benchmark over >5000 runs, as shown in the below figures.
> > >  https://imglink.cc/cdn/6kggrbSU-o.png, https://imglink.cc/cdn/EYWcrDB-sV.png, https://imglink.cc/cdn/vi0rdf4PKl.png
> > >
> > > **Training and inference time benchmarks:**
> > > Clarifies the scaling of network training time, showing that times remain tractable at higher dimension. We additionally included a new section which explicitly connects this scaling to the scaling of the underlying basis of the tree algebras.
> > > | Hopf Algebra | Metric | 2 | 4 | 8 | 12 | 16 |
> > > |-|-|-|-|-|-|-|
> > > | GL | Train Time | 23.68 s | 25.26 s | 41.54 s | 3.06 min | 4.20 min |
> > > | GL | Inference Time | 138.36 ms | 144.57 ms | 151.55 ms | 160.82 ms | 177.54 ms |
> > > | MKW | Train Time | 23.60 s | 25.53 s | 40.58 s | 3.11 min | 4.19 min |
> > > | MKW | Inference Time | 142.80 ms | 144.73 ms | 150.38 ms | 176.65 ms | 195.98 ms |
> > > | Shuffle | Train Time | 23.70 s | 24.89 s | 57.16 s | 2.80 min | 8.99 min |
> > > | Shuffle | Inference Time | 139.56 ms | 134.28 ms | 146.89 ms | 164.66 ms | 160.82 ms |
> > >
> > > **Euclidean timeseries benchmarks:**
> > > We undertook further testing on PPG-DaLiA long timeseries tasks, confirming identical performance to log-NCDE [Walker, et al. (2024)](https://arxiv.org/abs/2402.18512). As log-NCDE is exactly a special case of BNRDE, and performance is shown to be identical, their Euclidean long-timeseries and irregular timeseries results are directly applicable to our model.
> > >
> > > ## Sensitivity analysis
> > > **Depth sensitivity analysis:**
> > > New depth sensitivity analysis for rBergomi, and SO(3) experiments. This confirms depth 2 is sufficient and thus supports our claim that higher depths is not a practical blocker, as depth 2 is sufficient.
> > >
> > > **rBergomi experiment**
> > > | Depth | 128 | 256 | 384 | 512 |
> > > |-|-:|-:|-:|-:|
> > > | 2 | **0.06 ± 0.02** | **0.06 ± 0.01** | **0.07 ± 0.01** | **0.08 ± 0.02** |
> > > | 3 | 0.07 ± 0.01 | 0.06 ± 0.01 | 0.09 ± 0.01 | 0.10 ± 0.00 |
> > >
> > > **SO(3) experiment**
> > > | Depth | Pretraining Frobenius | Static Motion | Translation motion | Unconstrained motion |
> > > |---|---:|---:|---:|---:|
> > > | 2 | **0.049** | **_3.23_** | **_3.70_** | **_3.33_** |
> > > | 3 | 0.052 | _3.33_ | _4.22_ | _4.06_ |
> > >
> > > ## Ablations
> > > **rBergomi experiment:**
> > > New ablation confirms the branched kernel is a substantial contributor to performance, BNRDE additionally retains leading performance over RNN/stacked RNN (LSTM) models, excluded due to character limitations.
> > > | Method | Params | 128 | 256 | 384 | 512 |
> > > |---|---:|---:|---:|---:|---:|
> > > | BNRDE (GK) | 926 | 10.36 ± 1.35 | _8.53 ± 1.25_ | _7.94 ± 1.06_ | **5.84 ± 1.62** |
> > > | **BNRDE (BK)** | 927 | **6.50 ± 1.97** | **5.56 ± 1.35** | **7.14 ± 0.73** | _7.91 ± 1.63_ |
> > >
> > > We see similar results for the SPD experiment (S4.4), but we cannot show here due to character limits.

---

### Decision · Program_Chairs · 2026-04-30

**Decision:**

Accept (regular)

**Comment:**

This paper introduces Branched Neural Rough Differential Equations (B-NRDE), extending neural rough differential equations to handle Itô stochastic dynamics and manifold-valued systems by replacing standard geometric signatures with tree-based rough path lifts derived from Hopf algebras. The paper received consistently positive evaluations, with all reviewers acknowledging the mathematical soundness, clear problem motivation, and novelty of integrating branched rough paths into the neural differential equation framework. The primary concerns centered on scalability due to the combinatorial growth of tree-based representations, limited experimental breadth beyond manifold-constrained settings, and the tangent-space approximation error over long horizons. The paper makes a clear, well-executed advance in enabling geometrically and stochastically consistent neural dynamics modeling.